

# A conceptual model of oceanic heat transport in the Snowball Earth scenario

**D. Comeau[1], D. A. Kurtze[2], and J. M. Restrepo[3,4]**

[1]Center for Atmosphere Ocean Science, Courant Institute of Mathematical Sciences, New York University, NY, USA
[2]Department of Physics, Saint Joseph's University, Philadelphia, PA, USA
[3]Department of Mathematics, Oregon State University, Corvallis, OR, USA
[4]College of Earth, Ocean, and Atmospheric Sciences, Oregon State University, Corvalis, OR, USA

Received: 17 November 2015 – Accepted: 8 December 2015 – Published: 14 January 2016

Correspondence to: D. Comeau (comeau@cims.nyu.edu)

Published by Copernicus Publications on behalf of the European Geosciences Union.

**ESDD**

doi:10.5194/esd-2015-84

**Ocean heat transport in Snowball Earth scenario**

D. Comeau et al.

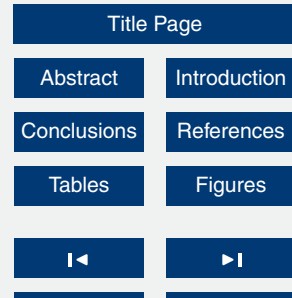

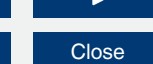

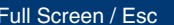



**ESDD**

doi:10.5194/esd-2015-84

**Ocean heat transport in Snowball Earth scenario**

D. Comeau et al.

## Abstract

Geologic evidence suggests that the Earth may have been completely covered in ice in the distant past, a state known as Snowball Earth. This is still the subject of controversy, and has been the focus of modeling work from low dimensional models up to state of the art general circulation models. In our present global climate, the ocean plays a large role in redistributing heat from the equatorial regions to high latitudes, and as an important part of the global heat budget, its role in the initiation a Snowball Earth, and the subsequent climate, is of great interest. To better understand the role of oceanic heat transport in the initiation of Snowball Earth, and the resulting global ice covered climate state, the goal of this inquiry is two-fold: we wish to propose the least complex model that can capture the Snowball scenario as well as the present day climate with partial ice cover, and we want to determine the relative importance of oceanic heat transport. To do this, we develop a simple model, incorporating thermohaline dynamics from traditional box ocean models, a radiative balance from energy balance models, as well as the more contemporary "sea glacier" model to account for viscous flow effects of extremely thick sea ice. The resulting model, consisting of dynamic ocean and ice components, is able to reproduce both Snowball Earth as well as present day conditions through reasonable changes in forcing parameters. We find that including or neglecting oceanic heat transport may lead to vastly different global climate states, and also that the parameterization of under ice heat transfer in the ice/ocean coupling, plays a key role in the resulting global climate state, demonstrating the regulatory effect of dynamic ocean heat transport. Furthermore we find that the ocean circulation direction exhibits bistability in the Snowball Earth regime.

## 1  Introduction

It is well known that the albedo difference between sea water and sea ice leads to a crucial climatic feedback. In a warming climate, melting of high albedo sea ice

**ESDD**

doi:10.5194/esd-2015-84

**Ocean heat transport in Snowball Earth scenario**

D. Comeau et al.

exposes low albedo sea water, thus increasing the fraction of incoming solar radiation that is absorbed and thereby amplifying warming. Conversely, in a cooling climate the growth of sea ice increases the planetary albedo, which amplifies cooling. Classic energy balance models (EBM)s demonstrate how this well-known ice–albedo feedback

can lead to multiple steady climate states (Budyko, 1969; Sellers, 1969). Given forcings which resemble present-day conditions, these models are bistable, with one possible steady state having a partial ice cover and another being completely ice free. With a sufficient reduction in the solar input or the greenhouse effect these energy balance models yield completely ice-covered steady states, reminiscent of the "Snowball Earth"

episodes of the Neoproterozoic era.

The virtue of simple models, of course, is that they make it possible to explore ranges of relevant parameters easily. First order energy balances at climatic scales are overwhelmingly radiative. The first EBMs were not intended to model meridional energy transport, and if included, it was incorporated as a diffusive process. Diffusion-

15 dominated transport tends to mitigate the tendency for sea ice to grow in a cooling climate: formation of extra sea ice would increase the temperature contrast between the ice-covered high latitudes and the low latitudes, which in turn would increase the rate of heat transport to the high latitudes. This then slows the growth of the ice, and exports some of the excess cooling due to the ice–albedo feedback to lower latitudes.

While the small ice cap instability has been a common phenomenon in EBMs, this bistability has not been seen in more sophisticated general circulation models (GCMs) (Armour et al., 2011). In a recent study, Wagner and Eisenman (2015) showed the bistability does not appear when an EBM was combined with a single column model for ice thermodynamics, suggesting sufficient complexity, including a seasonal cycle and

diffusive heat transport, eliminates this bifurcation. While the small ice cap instability is not the primary focus of this study, we will use our model to investigate this small ice cap bifurcation.

Another class of simple models, box ocean models, have been critical to understanding meridional ocean circulation, upwelling and mixing, among other

processes (Stommel, 1961). In particular, they have suggested important climate questions to pursue by means of more sophisticated models, data, or both. Models and measurements established, however, the importance of the basic thermohaline circulation and the more complete meridional ocean circulation. With it, models for
energy transport via diffusive processes were replaced by a more heterogeneous and dynamic mechanism. The Atlantic Meridional Overturning Circulation (AMOC), in particular, contributes a net energy transfer into the north Atlantic equivalent to several percent of the total incoming shortwave solar radiation incident to the region. Models of the meridional overturning circulation, from the simplest to the most detailed,
agree that under present-day conditions the circulation is bistable (Rahmstorf, 2000; Rahmstorf et al., 2005). In addition to the thermally-dominated steady state that is currently observed, a second, salinity-dominate steady state is also possible, with a much weaker circulation which flows in the direction opposite the present flow. It is clearly important to understand the interaction between this oceanic circulation and
the distribution of sea ice.

Since the present-day AMOC transports heat into the north Atlantic, it tends to reduce the extent of sea ice. A strengthening of the circulation would then reduce ice cover, while a weakening would cause it to expand. The circulation itself is driven by wind stress in and near the Southern Ocean, and it is sustained by variations in the
density of circulating water as it exchanges head and fresh water with the atmosphere as it flows along the surface. As water flows northward along the surface through the tropics it is warmed, and its salinity increases as a result of excess evaporation. The increase in temperature decreases the water density, while the increase in salinity increases it. As the water passes to higher latitudes it cools, and freshens
as precipitation exceeds evaporation. Again, the two effects tend to change the density in opposite directions. In the present configuration of the AMOC, the thermal effects dominate, so the water becomes denser as it moves through the subpolar latitudes, and ultimately sinks to return southward at depth.

**ESDD**

doi:10.5194/esd-2015-84

**Ocean heat transport in Snowball Earth scenario**

D. Comeau et al.

The presence of sea ice affects the processes that change the density of circulating sea water at high latitudes. An ice cover isolates the water from the atmosphere and so cuts off the precipitation that otherwise would reduce the salinity of the water and lower the rate at which its density increases. It also insulates the water thermally from the atmosphere. This by itself would not have much effect on the water density, since the water temperature cannot fall below the freezing temperature anyways. However, it allows the atmosphere to become colder than it would be if it were in contact with the sea water. Heat transfer to the cold atmosphere through the ice layer results in freezing of sea water at the base of the ice. Brine rejection then increases the salinity and hence the density of the remaining water.

A simple model of ocean circulation with sea ice may be useful in studying the Snowball Earth episodes in the Neoproterozoic era. Since the discovery of geologic evidence suggesting that glaciation occurred in the tropics at least twice in the Neoproterozoic (Kirschvink, 1992; Hoffman and Schrag, 2002), some 710 and 635 million years ago (Ma), there has been substantial debate about whether the Earth was ever in a completely ice-covered Snowball Earth state (Lubick, 2002). These events are thought to have lasted several million years, raising such questions as how life could have survived a long period if the Earth were in a completely ice-covered state (McKay, 2000). Thus in trying to explain these signs of apparent tropical glaciation in the context of global climate dynamics, alternative hypotheses have been proposed that leave some portion of the ocean either free of ice, or covered only in thin ice (Hyde et al., 2000; Pollard and Kasting, 2005; Abbot and Pierrehumbert, 2010; Abbot et al., 2011).

Poulsen et al. (2001) counts among the first studies to suggest dynamic ocean heat transport is important to the Snowball Earth hypothesis. Using the fully coupled Fast Ocean–Atmosphere Model (FOAM), initialized to Neoproterozoic parameter values to facilitate Snowball conditions, the authors found that a global ice cover was produced when using a mixed-layer ocean model that parameterized heat transport through diffusion. In fully coupled experiments with the ocean component, on the other hand,

Discussion Paper | Discussion Paper | Discussion Paper | Discussion Paper |

**ESDD**

doi:10.5194/esd-2015-84

**Ocean heat transport in Snowball Earth scenario**

D. Comeau et al.

the ice margin would retreat to high latitudes. Other studies have considered the Snowball Earth problem in an EBM framework, with ocean heat transport typically parameterized by a diffusion process (Pollard and Kasting, 2005; Rose and Marshall, 2009). In particular, Pollard and Kasting (2005) examine the feasibility of a tropical thin-ice solution, incorporating detailed treatment of optical properties of ice and a non-linear internal ice temperature profile, as well as a separate snow layer and an evaporation minus precipitation term to facilitate surface melt/accumulation.

GCMs that have more detailed ocean physics have also been used to study the initiation of a Snowball Earth (Yang et al., 2012a, b). A recent study by Voigt et al. (2011) uses the state-of-the-art atmosphere-ocean model ECHAM5/MPI-OM to study the Snowball Earth scenario. They implement a Marinoan (635 Ma) land mask in their coupled GCM simulations, as well as the lower insolation of a younger, weaker sun. In addition to ocean dynamics, their study also included sea ice dynamics (albeit with thin ice) and interactive clouds. All three had previously been found to be essential for Snowball initiation (Poulsen et al., 2001; Poulsen and Jacob, 2004; Lewis et al., 2003, 2007). Voigt et al. (2011) were able to achieve Snowball initiation, and also to prevent Snowball initiation in the same setting by doubling carbon dioxide levels. Stability analysis of an EBM analog, based on the 0-D model of global mean ocean temperature developed in Voigt and Marotzke (2010), indicates an insolation bifurcation point for Snowball Earth in the Marinoan setting of about 95–96 % of pre-industrial levels, in agreement with their computational results. In their experiments that resulted in partial ice cover, the ice margin was around 30–40° latitude, with maximum stable sea ice extent of 55 % of ocean cover observed in their experiments.

Sea ice in a global ice cover can be very thick, to the extent that flow by plastic deformation under its own weight should be considered. Thus its non-Newtonian fluid dynamics must be considered in addition to its thermodynamics. Goodman and Pierrehumbert (2003) (henceforth GP03) first considered these flow effects in the Snowball Earth scenario. (The same framework was used in Abbot and Pierrehumbert (2010) and Li and Pierrehumbert (2011) to transport dust to low latitudes in the Mudball

**ESDD**

doi:10.5194/esd-2015-84

**Ocean heat transport in Snowball Earth scenario**

D. Comeau et al.

Discussion Paper | Discussion Paper | Discussion Paper | Discussion Paper |

**ESDD**

doi:10.5194/esd-2015-84

**Ocean heat transport in Snowball Earth scenario**

D. Comeau et al.

scenario). Their model runs outside a global circulation model, using FOAM output for forcing data, and it has neither an active ocean component nor a parameterization for oceanic heat transport. They use the term "sea glacier" to describe their modeled ice, to distinguish it from present-day sea ice, which only grows to thicknesses on the order

of meters, and from land ice or ice shelves. The sea glacier is formed in the ocean, yet it achieves the thickness of a land ice sheet without the land–ice interface, and its non-Newtonian rheology is taken into account in the calculation of its flow. They are able to achieve both partial glaciation and a full Snowball state through changes in the atmospheric forcing (surface temperature and precipitation minus evaporation). They

find that the additional viscous flow term is highly effective at allowing the ice margin to penetrate low-latitude regions of melting, thus encouraging Snowball Earth initiation.

A recent study by Ashkenazy et al. (2013) found a dynamic ocean in a Snowball Earth scenario with strong circulation, in contrast to a stagnant ocean typically expected due to ice cover serving as an insulation layer to atmospheric forcing. Their

model was forced with geothermal heat, which was spatially varying with a peak near the Equator, averaging to $0.1\,\mathrm{W\,m^{-2}}$. In their 2-D and 3-D ocean simulations coupled with a 1-D ice model extending upon that of GP03, they found that the ocean plays a larger role in determining ice thickness than the atmosphere, and that geothermal heat forcing plays a dominant role in ice-covered ocean dynamics. This was expanded

upon in Ashkenazy et al. (2014), where a dynamic ocean with strong equatorial jets and a strong overturning circulation was found in simulations of a steady-state globally glaciated Earth.

Atmospheric dynamics and cloud cover undoubtedly play a large role in such climate systems, as demonstrated by Voigt et al. (2011). It is, however, difficult to isolate the

25 role played by oceanic transport in these coupled simulations, due to the necessary inclusion of the complex dynamics of the atmosphere and cloud distribution as well as sea ice dynamics. In fact, in Voigt and Abbot (2012) it was found that ocean heat transport has no effect on the critical sea ice cover that leads to Snowball initiation. This motivated us to consider a simpler model that includes oceanic heat transport coupled

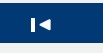 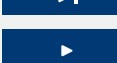

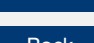 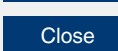

**ESDD**

doi:10.5194/esd-2015-84

**Ocean heat transport in Snowball Earth scenario**

D. Comeau et al.

to ice dynamics. We aim to extend the framework laid out in GP03 to include ocean heat transport effects, including under the ice layer. Realistic oceanic transport undoubtedly leads to highly non-uniform heat distributions, likely with local consequences on the global Snowball scenario. However, these local effects are beyond the scope of our study, and indeed beyond the scope of any low-dimensional model. By omitting atmospheric effects, we aim to get an assessment of the effects from oceanic heat transport alone.

The purpose of this paper is to investigate the interaction between sea ice and the meridional overturning circulation. By now there are several studies that have used complex circulation models to confirm that ocean transport is an important component of any explanation of how sea ice recedes and grows along with changes in forcing, albedo, biogeochemstry, etc. With a simple model, however, it is possible to efficiently test our understanding and propose questions critical to our being able to further understand the complexities of radiation, ice cover, and oceanic/atmospheric transport, such as the Snowball hypothesis.

To this end we have combined a one-dimensional energy balance model with a box model of the meridional overturning circulation and a dynamic ice component. Our model is described in detail in Sect. 2, a key element of which is the under ice heat exchange with the ocean. In Sect. 3 we present model results in different climate regimes. The sensitivity of the model to key parameterizations is studied in Sect. 4, and concluding remarks are in Sect. 5.

## 2 Model description

Our model consists of a four-box ocean model with transport, similar to that first proposed by Stommel (1961), coupled to a one-dimensional EBM similar to that of Budyko (1969) and a dynamic model for sea ice coverage (GP03), and is depicted in Fig. 1. The ocean component is a hemispheric model with thermohaline dynamics. While the ocean model uses a traditional transport equation for the salinity, it differs

from traditional box models in the use of an energy conservation model to capture the temperature dynamics. This allows us to couple the ice and radiation components to the ocean dynamics. The ice layer is zonally averaged, so its thickness is taken to depend only on colatitude $\theta$ and time $t$. The ice margin evolves dynamically, and we include non-Newtonian flow so that the model can accommodate an ice layer thick enough to be appropriate to Snowball conditions. The surface absorbs incoming solar radiation, with an albedo which takes into account whether the surface is open ocean or ice, and emits long-wave radiation with a specified emissivity. Where ice overlies the ocean, heat conducts through the ice layer and is exchanged with the ocean at the ocean–ice interface, where melting and freezing can occur, while also supplying heat for redistribution through ocean circulation, a departure from typical EBMs. We also account for geothermal heat forcing, found to be the dominant forcing in a Snowball Earth ocean (Ashkenazy et al., 2013).

In addition to physical properties of ice and sea water, geometrical features of the ocean basin, and the distribution of insolation, there are three parameters which are important to our studies. In the energy balance model we model the greenhouse effective by including an effective emissivity, $\varepsilon$, in terms of modeling outgoing longwave radiation. The box model requires a hydraulic coefficient, $k$, which relates the strength of circulation to the densities of the water in the various boxes. The third parameter quantifies the thermal coupling between the sea ice layer and the water beneath; we express this as an effective thermal boundary layer thickness, $D$, with the rate of heat transfer from the water to the ice being proportional to the temperature difference between the box water and the base of the ice, divided by the thickness $D$. Among the issues we will investigate is the question of how, and indeed whether, the coupling relates the bistability of the energy balance model and the independent bistability of the box model.

**ESDD**

doi:10.5194/esd-2015-84

**Ocean heat transport in Snowball Earth scenario**

D. Comeau et al.

**ESDD**

doi:10.5194/esd-2015-84

**Ocean heat transport in Snowball Earth scenario**

D. Comeau et al.

## 2.1 Ocean and energy components

The ocean component of our model is similar to the one proposed by Griffies and Tziperman (1995) and Kurtze et al. (2010). Four boxes are used to represent the ocean in one hemisphere, from pole to Equator, with each box representing a zonal average across longitude. Referring to Fig. 1, we define Box 1 as the tropical surface ocean box, Box 2 its polar counterpart, Box 3 below Box 2, and Box 4 below Box 1. The depth of the upper boxes is $d_u$ and the depth of the lower boxes is $d_l$, with $d_u \ll d_l$. Boxes 2 and 3 extend from the pole at colatitude $\theta = 0°$ to a fixed boundary at $\theta = \zeta$. Boxes 1 and 4 extend from $\theta = \zeta$ to the Equator at $\theta = 90°$. We choose the colatitude boundary $\zeta$ to be $45°$ simply because we are interested in investigating climate regimes ranging from global ice cover to zero ice cover, so there is no advantage in trying to confine the ice cover to a polar box only. We have found changing the location of this boundary within midlatitudes does not qualitatively affect our results. The dynamic ice margin is at $\eta(t)$, the ice cover thickness is given by $h(\theta, t)$, and its poleward meridional velocity is given by $v(\theta, t)$. Model parameter values, including geometric quantities pertaining to the box structure, are given in Table 2.

Each box has a (well-mixed) temperature $T_j(t)$ and salinity $S_j(t)$, $j = 1, \ldots, 4$, which determine the density of each box by a linear equation of state

$$\rho_j(T_j, S_j) = \rho_0 \left[ 1 + \beta_S(S_j - S_0) - \beta_T(T_j - T_0) \right].$$

Here $\rho_0$ is a reference density corresponding to a reference temperature and salinity $T_0$, $S_0$. $\beta_S$ and $\beta_T$ are the expansion coefficients associated with salinity and temperature. The density-driven flow between the boxes is denoted by $f$, where we adopt the convention that $f < 0$ is surface poleward flow (from Box 1 to Box 2). As in Griffies and Tziperman (1995), the (buoyancy driven) transport rate is

$$f = k \left[ \frac{d_u}{d_l}(\rho_1 - \rho_2) + (\rho_4 - \rho_3) \right], \tag{1}$$

Discussion Paper | Discussion Paper | Discussion Paper | Discussion Paper |

where $k = k_0 = 8 \times 10^4 / \rho_0$ Sv is the hydraulic constant which governs the strength of the density driven flow. In Sect. 4.2 we explore the model's sensitivity to this parameter. The flux $f$ is purely thermohaline-driven. We could modify this flux to include the effect of wind stresses in a crude manner by an additive correction to the flux, however, this has been omitted in this study.

The equations for each box's salinity and temperature will depend on the direction of the mean meridional flow. The salinity equations when $f < 0$, corresponding to poleward surface flow, are:

$$V_1 \frac{\mathrm{d}S_1}{\mathrm{d}t} = |f|(S_4 - S_1) + S_1 \int_{\zeta}^{\max \zeta, \eta} M(\theta) r_E^2 \sin \theta \mathrm{d}\theta,$$

$$V_2 \frac{\mathrm{d}S_2}{\mathrm{d}t} = |f|(S_1 - S_2) + S_2 \int_{0}^{\min \zeta, \eta} M(\theta) r_E^2 \sin \theta \mathrm{d}\theta,$$

$$V_3 \frac{\mathrm{d}S_3}{\mathrm{d}t} = |f|(S_2 - S_3),$$

$$V_4 \frac{\mathrm{d}S_4}{\mathrm{d}t} = |f|(S_3 - S_4). \tag{2}$$

Here $V_j$ is the volume of the $j$th box, and $M(\theta, t)$ is the total production/melting rate of ice, with $M > 0$ corresponding to ice production and $M < 0$ corresponding to melting, described in Sect. 2.2. The terms involving the circulation rate $f$ correspond to fluxes across the box boundaries. We assume that ice sits atop the surface ocean boxes, and that the mass of ice is much less than the total mass of ocean water. The surface ocean box volumes $V_j$ are kept constant, and any changes to the deep ocean box volumes are negligible. An important assumption is that ice that is formed is freshwater ice, and as such rejects brine into the ocean. The integral term represents the change in salinity due to net freshwater added/removed through ice melting/production. The bounds of

Discussion Paper | Discussion Paper | Discussion Paper | Discussion Paper

**ESDD**

doi:10.5194/esd-2015-84

**Ocean heat transport in Snowball Earth scenario**

D. Comeau et al.

integration represent the portion of each box covered in ice, and $r_E$ is the Earth's radius. The ice component of the model serves as a saline forcing on the ocean box model component. There are similar equations for when ocean circulation is in the reverse direction.

In contrast to traditional Stommel box models, rather than using transport equations for the temperature we opt instead for thermal balance equations. The box temperatures change as heat transfers between boxes with the flow $f$, and as it transfers into the box via net radiation or conduction through overlying ice. Thus for $f < 0$ the temperature equations are

$$c_w V_1 \frac{d(\rho_1 T_1)}{dt} = c_w |f| (\rho_4 T_4 - \rho_1 T_1) + \int_{\max \zeta, \eta}^{\pi/2} \left( (1 - \alpha_w) F_s(\theta) - \varepsilon \sigma T_1^4 \right) r_E^2 \sin \theta \, d\theta$$

$$- \frac{\kappa_w}{D} (T_1 - T_f) \int_{\zeta}^{\max \zeta, \eta} r_E^2 \sin \theta \, d\theta, \tag{3}$$

$$c_w V_2 \frac{d(\rho_2 T_2)}{dt} = c_w |f| (\rho_1 T_1 - \rho_2 T_2) + \int_{\min \zeta, \eta}^{\zeta} \left( (1 - \alpha_w) F_s(\theta) - \varepsilon \sigma T_2^4 \right) r_E^2 \sin \theta \, d\theta$$

$$- \frac{\kappa_w}{D} (T_2 - T_f) \int_{0}^{\min \zeta, \eta} r_E^2 \sin \theta \, d\theta, \tag{4}$$

$$c_w V_3 \frac{d(\rho_3 T_3)}{dt} = c_w |f| (\rho_2 T_2 - \rho_3 T_3) + F_g \int_{0}^{\zeta} r_E^2 \sin \theta \, d\theta, \tag{5}$$

Discussion Paper | Discussion Paper | Discussion Paper | Discussion Paper | Discussion Paper |

ESDD

doi:10.5194/esd-2015-84

Ocean heat transport in Snowball Earth scenario

D. Comeau et al.



$$c_w V_4 \frac{\mathrm{d}(\rho_4 T_4)}{\mathrm{d}t} = c_w |f| (\rho_3 T_3 - \rho_4 T_4) + F_g \int_\zeta^{\pi/2} r_E^2 \sin\theta \mathrm{d}\theta. \qquad (6)$$

Here $c_w$ is the ocean water heat capacity. The first term in each equation accounts for net energy accumulation due to fluxes across box boundaries. The second term in Eqs. (3) and (4) represents the radiative balance, in a similar form as appears in EBMs dating back to Budyko (1969). Here $\alpha_w$ is the ocean water albedo, and $F_s(\theta)$ is insolation, for which we use the parameterization of McGehee and Lehman (2012):

$$F_s(\theta) = \frac{342.95}{\sqrt{1-e^2}} \frac{2}{\pi^2} \int_0^{2\pi} [1 - (\cos\theta \sin\beta \cos\gamma - \sin\theta \cos\beta)^2]^{1/2} \mathrm{d}\gamma. \qquad (7)$$

Here $e$ is the eccentricity of the Earth's orbit (presently at 0.0167), $\beta$ is the obliquity (presently at 23.5°), and $\gamma$ is longitude. This parameterization is annually averaged (no seasonal cycle), but with time-dependent orbital parameters that allows for accounting for the Milankovitch cycles. However, in our results we found these were not strong enough to qualitatively affect the resulting model state, regarding the location of the ice margin, or Snowball Earth initiation or deglaciation, so only present day insolation values are used. The final terms in Eqs. (5) and (6) represent a uniform geothermal heating forcing of $F_g = 0.05\,\mathrm{W\,m^{-2}}$, as in Ashkenazy et al. (2013).

In Eqs. (3) and (4), insolation is balanced by outgoing long wave radiation, integrated over the exposed ocean portion of the box, where we assume blackbody radiation from the surface using the full Stefan–Boltzmann law, with $\sigma$ the Stefan–Boltzmann constant. As mentioned, $\varepsilon$ is the effective emissivity, which is the ratio of outgoing longwave radiation emitted at the top-of-atmosphere to that emitted at the Earth's surface, and therefore represents the greenhouse effect. Thus atmospheric effects are distilled into this single parameter, which we will use as our control between climate states.

**ESDD**

doi:10.5194/esd-2015-84

**Ocean heat transport in Snowball Earth scenario**

D. Comeau et al.

The last terms in Eqs. (3) and (4) represent ice/ocean coupling by modeling heat transfer between the ocean box and its ice cover, a key component of our model. This includes the parameter $D$, having units of length, which parameterizes the under ice exchange of energy with the ocean and we refer to as the effective thermal boundary layer. We note that $D$ is a key unconstrained parameter, which by default we set to $D = 0.05$ m, and explore the model's sensitivity to this parameter in Sect. 4. Here $\kappa_w$ is the sea water thermal conductivity, which we take to be constant, neglecting dependence on salinity and temperature.

The ocean heat transport in this model is then quantified as

$$H_{ocean} = f c_w \rho_w (T_1 - T_2).$$

## 2.2 Sea ice component

We largely follow the "sea glacier" treatment of GP03 and later Li and Pierrehumbert (2011), with a few noted exceptions. The details of the rheology of sea glaciers are left to the Appendix A. The equation for the ice thickness, by conservation of mass, is

$$\frac{\partial h(\theta,t)}{\partial t} + \nabla \cdot [v(\theta,t)h(\theta,t)] = M(\theta,t), \tag{8}$$

where $h(\theta,t)$ is ice thickness, $v(\theta,t)$ is meridional ice velocity, and $M(\theta,t)$ is the ice melting/production term. The equation for $v$ is given by a Glen's flow law (GP03),

$$\nabla \cdot v(\theta,t) = \mu^n h(\theta,t)^n, \tag{9}$$

where $\mu$ is a temperature dependent viscocity parameter accounting for the non-Newtonian rheology of the ice, the details of which are left to the Appendix A.

The ice melting/accumulation term $M(\theta,t)$ is a departure from the treatment of GP03. Ice melting or production can occur either from heat transferred through the ice from the surface, or from heat transferred through the ocean through the effective ice/ocean

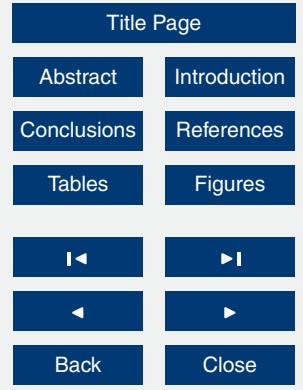

**ESDD**

doi:10.5194/esd-2015-84

**Ocean heat transport in Snowball Earth scenario**

D. Comeau et al.

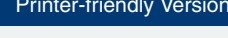

Interactive Discussion

thermal boundary layer $D$, and is given by

$$LM(\theta,t) = \frac{\kappa_i}{\rho_i}\frac{T_f - T_s(\theta,t)}{h(\theta,t)} - \frac{\kappa_w}{\rho_w}\frac{T_{1,2}(\theta,t) - T_f}{D},\qquad(10)$$

where $L$ is the latent heat of fusion of ice. When $M(\theta,t) > 0$, there is net accumulation of ice, and when $M(\theta,t) < 0$, there is net melting. The first term on the right side accounts for heat transfer through the ice, assuming a linear temperature profile in the ice from the surface $T_s(\theta)$ to the base at freezing $T_f$. The second term accounts for heat transfer with the ocean through the parameter $D$; an equivalent term appears in the energy budget for the ocean box temperatures in Eqs. (3) and (4). Since only average ocean box temperatures are computed by our model, to prevent an artificial and arbitrary jump in temperature across the box boundary from influencing the melting term, the step function surface temperature profile $T_1(t)$, $T_2(t)$ is regularized to a smooth $T_{1,2}(\theta,t)$ for use in Eq. (10). Note that our model only accounts for melting and freezing at the base of the ice, and there are no terms that model melting at the upper surface or accumulation due to evaporation/precipitation forcing.

The ice surface temperature $T_s(\theta,t)$ is given by a primary radiative balance, as well as a term accounting for heat transfer through the ice. The (average annual) ice surface temperature $T_s(\theta)$ is given by

$$c_i \rho_i h(\theta)\frac{dT_s(\theta)}{dt} = F_s(\theta)(1 - \alpha_i) - \varepsilon\sigma T_s(\theta)^4 + \kappa_i\frac{T_f - T_s(\theta)}{h(\theta)},\qquad(11)$$

where $c_i$ is the specific heat, $\alpha_i$ the albedo, and $\kappa_i$ the thermal conductivity of ice. The last term of Eq. (11) accounts for heat transfer through the ice, as in Eq. (10).

## 2.3 Model setup

Settings for the parameters are listed in Table 2. The ocean component is run on a yearly time step, with the ice dynamics sub-cycled on a monthly time step. (This

**ESDD**

doi:10.5194/esd-2015-84

**Ocean heat transport in Snowball Earth scenario**

D. Comeau et al.

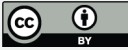

does not imply a seasonal cycle; rather we keep the insolation constant, as given in Eq. 7). For each ocean time step, we solve the set of differential equations for box temperatures and salinities (Eqs. 2–6), and ice surface temperature (Eq. 11) using a simple forward Euler method. At each ice time step, we solve (Eq. 8) using

a second-order upwind scheme after solving for the velocity in (Eq. 9). We discretize our longitudinal domain with 100 points, solving for ice thickness $h$ and velocity $v$ on staggered grids, and set $v = 0$ at the pole for the boundary condition in (Eq. 9). We initialize the model in an ice free state, so any ice is formed through the model by Eq. (10). Following the setup of Griffies and Tziperman (1995), with initial box

temperatures $T_1 = 298\,K$, $T_2 = T_3 = T_4 = 273\,K$ and salinities $S_1 = 36.5$, $S_2 = 34.5$, and $S_3 = S_4 = 35\,psu$. Models typically reach an equilibrium climate steady state after 10 000 model years, and the following results are at the end of 20 000 model year runs.

## 3 Results

As mentioned, we use the effective emissivity $\varepsilon$ as a control between model climate states. We emphasize that, as we have no atmospheric heat transport or other atmospheric effects in our model other than this parameter to crudely account for greenhouse effects, we do not attribute physical meaning to this parameter value. With a choice of $\varepsilon = 0.6$, which is estimated to be a reasonable value for current climate

(Voigt et al., 2011), the model remains in an ice-free planet with a thermally driven poleward circulation of $\approx 29\,Sv$ and associated heat transport $\approx 0.81\,PW$. We note the circulation strength is in line with results referenced in Griffies and Tziperman (1995) that give an approximate meridional circulation strength of 20 Sv from the coupled ocean–atmosphere model of NOAA's Geophysical Fluid Dynamics Laboratory (GFDL)

model. The salinities of the boxes quickly mix and converge to roughly the same value of 35 psu. The equatorial surface box temperature settles to $T_1 = 288.5\,K$, whereas the other three boxes converge to a common temperature of $T_2 = T_3 = T_4 = 281.7\,K$.

**ESDD**

doi:10.5194/esd-2015-84

**Ocean heat transport in Snowball Earth scenario**

D. Comeau et al.

Discussion Paper | Discussion Paper | Discussion Paper | Discussion Paper |

**ESDD**

doi:10.5194/esd-2015-84

**Ocean heat transport in Snowball Earth scenario**

D. Comeau et al.

Hence we have a strong, thermally dominated poleward circulation in this simulation. Depending on the choice of the $\varepsilon$ parameter (which controls the radiative balance and parameterizes any atmospheric effects), the model can reproduce present day partial ice cover conditions, as well as Snowball Earth global ice cover conditions, as we will now describe.

## 3.1 Partial glaciation

Raising the effective emissivity from $\varepsilon = 0.6$ to $\varepsilon = 0.7$, we move to an equilibrium climate state with a small, stable ice cover. To determine the role of oceanic heat transport, we run the model with the circulation rate $f$ set to zero for comparison against the full model run. Figure 2a and b shows the ice thickness and ice velocity profiles with and without ocean circulation, and we see the ocean circulation is effective at reducing ice thickness as well as pulling the ice margin north. Without the additional heat from the equatorial region moving poleward, the polar region remains cool, facilitating ice growth.

The ice cover is approximately 500 m thick at the pole in the full model run, much thicker than current sea ice cover; however, this is consistent with partial glaciation results from GP03. It is because the ice is this thick that viscous flow effects need to be considered. There is a strong response in the ice velocity, largely due to the thicker ice cover when ocean circulation is not included, resulting in stronger viscous flow. The ice velocity reaches its maximum just before the ice margin (approximately 120 m yr$^{-1}$ in the full simulation). We also note the ice surface temperature seen in Fig. 2c, calculated by Eq. (11), is in line with the air surface temperature forcing used in the experiments of GP03. In Fig. 2d we see the accumulation term becoming negative near the ice margin, indiciating a region of net melting that stabilizes the ice margin.

The steady state ocean circulation strength in this partial ice cover scenario is $\approx$ 26.3 Sv, which is closer to the numerical results of the GFDL model referenced in Griffies and Tziperman (1995) than the ice-free run. As with the ice free runs, the box salinities quickly mix to the same value of approximately 35 psu, while the surface

**ESDD**

doi:10.5194/esd-2015-84

**Ocean heat transport
in Snowball Earth
scenario**

D. Comeau et al.

equatorial box temperature settles to $T_1 \approx 278.6\,\mathrm{K}$, and the other boxes mix to $T_2 \approx T_3 \approx T_4 \approx 272.4\,\mathrm{K}$. By increasing the effective emissivity $\varepsilon$, the model steady state ice profile moves smoothly futher equatorward, until a large ice cap instability threshold is reached. When ice appears south of this instability threshold, the entire planet is covered in ice and a Snowball Earth state is reached.

## 3.2 Global glaciation

To approximate the Neoproterozoic climate in our model, we lower insolation to 94 % of its current value, accounting for a weaker, younger Sun (Voigt et al., 2011). Raising the effective emissivity from $\varepsilon = 0.7$ to $\varepsilon = 0.85$, we move from a climate state with a small, stable ice cover to global ice cover, a Snowball Earth, as shown in Fig. 3. With this global ice cover scenario, there is no region of net ice melting, as seen in Fig. 3d, and without a manual change in forcing, such as an increase in the greenhouse effect until melting occurs, and ice will continue to accumulate without bound. However, the ice margin takes roughly 5000 years to reach the Equator from initialization, and then produces a relatively neglible amount of ice. With continual positive net ice production, our assumption of ice being completely salt free produces ever increasing and unphysical salinity levels in the ocean boxes. However, since we have neglected salinity dependence of the freezing temperature, the model only depends on the salinity gradients between boxes, so unphysical salt content does not affect other aspects of the model.

In this scenario, the ocean circulation reverses direction to a much weaker, saline-driven, equatorward circulation with a strength of approximately 4.9 Sv. Ocean circulation thus has a minor effect on the climate state, particularly on ice thickness and temperature profiles seen in Fig. 3. The ocean box temperatures largely mix to a common value just above ocean freezing temperature, and the resulting flow is driven by salinity gradients. The ice thickness profile in Fig. 3a shows a peak thickness at the pole of nearly 475 m, reducing to about 175 m near the Equator. The back-pressure term in the boundary condition for ice velocity in Eq. (A2) brings the ice velocity to

**ESDD**

doi:10.5194/esd-2015-84

**Ocean heat transport in Snowball Earth scenario**

D. Comeau et al.

Discussion Paper | Discussion Paper | Discussion Paper | Discussion Paper

zero at the Equator (Fig. 3b), the effect of which is to move the maximum velocity ($\approx 55\,\mathrm{m\,yr^{-1}}$) to a location in the midlatitudes.

Examining the model results between these very different climate states of a small, stable ice cap at $\varepsilon = 0.7$ and Snowball Earth at $\varepsilon = 0.85$, we find a threshold where the model transitions. With an effective emissivity of $\varepsilon = 0.84$, the equilibrium climate state is near the large ice cap instability threshold, and we get a strong response from the ocean circulation. In Fig. 4, we show the results of a simulation with and without ocean circulation dynamics. We observe in Fig. 4a, that without oceanic heat transport, we get a Snowball Earth, but with oceanic heat transport, the ice line is held back from the Equator. As with the previous Snowball Earth state, the ocean circulation strength here, 2.7 Sv (equatorward) is considerably weaker than in the small ice cap simulation. However even this weakened circulation, and thus weakened oceanic heat transport, is still enough to drive the climate into a Snowball state if turned off, demonstrating strong sensitivity in this regime.

## 4  Model sensitivity to ocean and atmosphere parameters

We have already seen the model sensitivity to the effective emissivity parameter $\varepsilon$, which we used to drive the model through vastly different climate states in Sect. 3. There are other key sensitivities the model exhibits, which we now discuss, namely in the parameterizion of energy transfer between the ocean and ice components, and the scaling strength of ocean circulation. By the nature of the large ice cap instability, the threshold effective emissivity of $\varepsilon = 0.84$ seen in Sect. 3.2 is sensitive to the parameter choices representing these key processes. We also explore the model's bistability in the small ice cap and large ice cap instabilities.

## 4.1 Ocean/ice energy transfer parameterization

The parameter $D$ is responsible for parameterizing the transfer of energy between the ocean and ice systems, a so called effective thermal boundary layer, and is the least constrained parameter in our model. It appears in equations for the surface ocean box temperatures (Eq. 3), (Eq. 4) as well as in ice melting/production (Eq. 10). The role of $D$ has competing effects in these two equations. For the ocean box temperatures, $D$ appears in the term corresponding to energy loss due to the presence of the ice cover, and thus increasing $D$ cools the ocean. However this energy loss is balanced in the system by the ice melting/production term, where increasing $D$ encourages melting. In the limiting case of $D = 0$, the ocean and basal ice would be forced to the same temperature, and increasing $D$ further insulates the ocean from the ice, by slowing the heat transfer between the two component. There are also other feedbacks in the system, notably the indirect effect of $D$ on the strength of the ocean circulation, and thereby oceanic heat transport, which we have already seen can strongly affect ice cover.

We explore the model's sensitivity to values of this parameter over two orders of magnitude in Fig. 5 in both the small ice cap regime ($\varepsilon = 0.75$), and near the large ice cap instability ($\varepsilon = 0.84$, insolation at 94 % current values), showing ice thickness profiles and ice melting/accummulation rate at the end of a 20 000 year run to equilibrium. In the small ice cap regime, increasing $D$ steadily reduces ice thickness, though the ice margin remains in a small high latitude range across changes in $D$, apart from the large end of the range, where the margin pushes further equatorward. The ocean circulation (poleward) also steadily reduces with increased $D$, from 20 Sv circulation with $D = 0.01$ down to 10 Sv circulation for $D = 1$. There is, however, a strong response in the ice melting/accumulation term, where large values of $D$ yield small magnitude melting terms. This proved to be critical in melting excess ice, which we saw in our experiments with changing solar forcing discussed below in Sect. 4.3.

Discussion Paper | Discussion Paper | Discussion Paper | Discussion Paper |

**ESDD**

doi:10.5194/esd-2015-84

**Ocean heat transport in Snowball Earth scenario**

D. Comeau et al.

The large ice cap threshold emissivity for $D = 0.05$ is $\varepsilon = 0.84$, and we see the instability reflected in that the other values of $D$ result in a global ice cover. There is an interesting divide in ocean circulation in the large ice cap regime, where the large tested values of $D = 0.5, 1$ result in $\approx 7.5\,\mathrm{Sv}$ poleward circulation, whereas the other $D$ values result in $\approx 3$–$5\,\mathrm{Sv}$ equatorward circulation. Note that in Fig. 5c, we see different behavior in the ice accumulation, and thus freshwater forcing, term associated with these different directions of circulation. The more spatially uniform accumulation terms, $D = 0.5, 1$ are associated with poleward circulation, and the other global ice accumulation terms share a similar shape increasing with colatitude. Thus we see both directions of ocean circulation are consistent with the Snowball Earth scenario (albiet only demonstrated with different parameters so far). The sensitivity of the model results to this simple parameterization of heat flux between the ocean and ice cover demonstrates the importance of this exchange in the heat budget.

## 4.2 Circulation constant

The default value of the circulation constant $k = k_0$ in (Eq. 1) from Griffies and Tziperman (1995) results in a circulation strength of $\approx 20\,\mathrm{Sv}$, in line with estimates of present-day mean meridional ocean circulation. This may not be appropriate for the Neoproterozoic, warranting an examination of the sensitivity of the model to changes to this parameter, explored in Fig. 6. We again show results for the same experimental setups as in Sect. 4.1. Here we see a stronger effect on the ice margin in the small ice cap regime, where smaller circulation constants reduce heat transport from the tropics to the poles, thereby facilitating ice growth (note circulation always remains poleward in this regime). In the large ice cap regime, we again see the sensitivity of the large ice cap instability, as deviating from the default circulation strength constant, in either direction moves the threshold emissivity away from $\varepsilon = 0.84$, and the resulting climate state is a Snowball Earth. For the Snowball Earth states, there is also an interesting divide in direction of circulation, in that all circulations are actually poleward, ranging from 16 to 1 Sv, except for the $k = 2 \times k_0$ case, where the Snowball Earth resulting circulation

**ESDD**

doi:10.5194/esd-2015-84

**Ocean heat transport in Snowball Earth scenario**

D. Comeau et al.

is equatorward at 7 Sv. In Fig. 6 we see the same association of accumulation rate profiles with circulation direction, where elongated S shape for $k = 2 \times k_0$ corresponds to equatorward circulation, and the other more spatially uniform profiles correspond to poleward circulation.

## 4.3 Model bistability

As discussed, a well-known feature of EBMs is hysteresis with respect to radiative forcing. To study this in our model, we setup experiments to investigate the hysteresis loop in the small ice cap instability and the large ice cap instability. In the small ice cap case, we began with conditions that resulted in a small ice cap, gradually increased radiative forcing by decreasing the emissivity $\varepsilon$ until the ice cap melted away, and then increased $\varepsilon$ to its starting value. With each incremental change in $\varepsilon$, the model is run for 10 000 years to ensure that the model is in equilibrium. A similar experiment was run for the large ice cap case, except with the forcing changes in cooling first, until Snowball Earth is reached, and then subsequent warming. The results of these experiments are shown in Fig. 7.

The expected hysteresis loops in each scenario manifest themselves in the left column of Fig. 7. In the small ice cap case, The ice margin begins with $\varepsilon = 0.7$ near $17°$ colatitude, and following the red path through decreasing $\varepsilon$, we see the ice cap abruptly change from a margin at $10°$ colatitude, to completely disappear at $\varepsilon = 0.6$. As the forcing is then cooled through the blue path, the model remains ice free until the appearance of a small ice cap near $\varepsilon = 0.68$, returning to its starting location at $\varepsilon = 0.7$. Looking at the response of the ocean circulation through the experiment, there is a small response in the strength of circulation in the hysteresis loop. As alluded to in Sect. 4.1, this behavior was actually used to constrain the value of $D = 0.05$, as larger values of $D$ did not yield adequate melting rates to melt ice once it appeared, resulting in very large hysteresis loops.

In Fig. 7c, we see the familiar large ice cap instability hysteresis loop, where as the climate is cooled through increasing $\varepsilon$, the ice margin gradually increases until

Discussion Paper | Discussion Paper | Discussion Paper | Discussion Paper | Discussion Paper

**ESDD**

doi:10.5194/esd-2015-84

**Ocean heat transport in Snowball Earth scenario**

D. Comeau et al.

**ESDD**

doi:10.5194/esd-2015-84

**Ocean heat transport in Snowball Earth scenario**

D. Comeau et al.

the large ice cap instability is reached near 70° colatitude, beyond which Snowball Earth is reached. Notable is that in this gradual cooling, the ocean circulation remains poleward through the large ice cap instability to Snowball Earth, the opposite direction that was seen for the same parameters and forcing in Sect. 3.2 when the model was initialized and run in the regime to produce large ice cap or Snowball conditions. Thus here we have true bistability of the direction of ocean circulation in the large ice cap and Snowball Earth regime. As the climate is subsequently warmed, Snowball Earth is maintained, but notably the ocean circulation remains poleward, continuining to transport heat to the poles, and in fact with a strong increase in warming (up to $\varepsilon = 0.4$), Snowball Earth is escaped in the model.

As there is also present concern of a collapse of the AMOC, we also wanted to see the effect of changing the ocean circulation forcing to weaken to zero, then increase again. For this we manually changed the circulation constant $k$ in Eq. (1), weakening $k$ from $k_0$ to 0, and then back to $k_0$. In Fig. 8, we see that while the resulting circulation follows a very close path through weakening and strengthening again, despite the ice margin not retreating during the strengthening phase. The more interesting case however is the large ice cap scenario, where weakening the circulation constant sends the model into a Snowball Earth, which is then inescapable as manually increasing the circulation constant only furthers the equatorward circulation.

# 5   Conclusions

We have presented a low dimensional conceptual climate model consistent with elements of classical low dimensional models that is able to reproduce both present day, partial ice cover climate, as well as a Snowball Earth global ice cover climate. The radiative balance terms similar to those in EBMs produce states of ice cover consistent with classical EBM results of two stable solutions, a small ice cover and a global ice cover, as well as an unstable solution, a large but finite ice extent. The ocean box model produces ocean states in two circulation regimes: a strong, poleward, thermally driven

circulation, as well as a weaker, equatorward, saline driven circulation. A summary of these results is given in Table 1.

Our primary interest is investigating the role of dynamic ocean circulation in the initiation of Snowball Earth, particularly in the large ice cap instability. We find in parameter regimes where there is no ice or a small ice cap, the ocean circulation is expectedly thermally dominated and poleward. Moving through parameter space to larger ice margins, the ocean circulation weakens, until ultimately reversing direction near the large ice cap instability, resulting in a Snowball Earth scenario and a weak, equatorward circulation. Discounting ocean circulation altogether allows for easier transition into global ice cover. We find, together with the effective emissivity which parameterizes the atmospheric component, that the energy transfer between the ice and ocean components plays a crucial role in determining the model's resulting climate state.

The results from our ice component are largely in line with those of GP03, in both ice thickness and position of the ice margin, despite some key differences in modeling approach. There is a notable difference in the ice velocity, however, in that our computed ice velocities are an order of magnitude less those reported in GP03. One possible reason for this is that the viscosity parameter $\mu$, which GP03 are only required to calculate once due to a static surface temperature forcing, is recalculated in our model in response to the dynamic surface temperature in (Eq. 11). Our steady state surface temperature (shown in Fig. 4) is cooler than the forcing used in GP03 partial glaciation case, and thus our cooler surface temperature creates more viscous ice, slowing down ice flow. The ice thickness profiles in our Snowball experiments vary smoothly by a couple hundred meters from pole to Equator, in contrast to sea glacier models that obtain a more uniform thickness in Snowball state, as in Li and Pierrehumbert (2011). Our results also qualitatively agree with that of Pollard and Kasting (2005, e.g. ice thickness, velocity, ice accumulation rates, surface temperature) in the steady state Snowball Earth scenario, with the notable exception of a sharper decline in tropical ice thickness, and while we are confident that adopting their more

Discussion Paper | Discussion Paper | Discussion Paper | Discussion Paper |

**ESDD**

doi:10.5194/esd-2015-84

**Ocean heat transport in Snowball Earth scenario**

D. Comeau et al.

developed ice model would not significantly change the results of our model presented here in the Snowball Earth scenario (apart from a thinner tropical ice), it would certainly be of interest to study transient behavior and parameter ranges that lead to climate regimes other than Snowball Earth.

The ocean in our Snowball scenario has a considerably weaker circulation strength of 4.9 Sv than present-day estimates, and while this is not indicative of a stagnant ocean, it is not as strong as the circulation strengths of approximately 35 Sv that were achieved in the study of Ashkenazy et al. (2013). Their 2-D simulations allowed for resolution of both vertical mixing and horizontal eddies, and while they also did not have land in their simulations, they did have an underwater ocean ridge that had a highly localized and higher strength geothermal heat forcing corresponding to a spreading center, which, together with strong vertical salinity profiles, drove the strong circulation.

The main conclusion we reach from this study is that ocean circulation and its associated heat transport play a vital role in determining the global climate state and ice cover. We have seen in the partial glaciation case that the ocean circulation severely inhibits ice growth, and we have seen in the near global glaciation case that even in that state's severely weakened ocean circulation, lack of oceanic transport leads to a drastically different Snowball Earth state. In particular, our study found that both heat flux between the ice and ocean, and the value of the circulation constant that controls circulation strength (tuned to present day conditions) played a crucial role in determining the global climate state. Not only do we find bistability in the small and large ice cap instability regimes with regards to radiative forcing, we also find that both thermally driven poleward, and saline driven equatorward circulations are consistent with a Snowball Earth scenario, a bistability arising from indirect freshwater forcing through the model's ice melting/accumulation term.

Our inclusion of a simple parameterization of under-ice ocean heat transport, mediating dynamic oceanic heat transfer not present in traditional EBM diffusive heat transport models is here found to be crucial in determining the steady state climate regime. This warrants further investigation in a GCM setting, to examine the role played

**ESDD**

doi:10.5194/esd-2015-84

**Ocean heat transport in Snowball Earth scenario**

D. Comeau et al.

by thermal processes in the ice and ocean that account for heat transport in the sub-ice cover layer. To be more direct, we find that oceanic heat transport is crucial to understanding Snowball initiation; that the ice cover affects it significantly; that the results are sensitive to the water–ice thermal coupling and the factors driving the circulation; and that it is therefore worthwhile to use GCMs to investigate these factors in detail.

While atmospheric effects were largely neglected for simplicity and because our focus was on the role of oceanic heat transport in the Snowball Earth setting, including an atmospheric component would be the natural progression of this work, particularly a precipitation–evaporation component that would facilitate ice surface melting/accumulation, though it is worth noting that we were able to reproduce both present day and Snowball Earth conditions without a precipitation–evaporation forcing.

## Appendix A: Sea glacier rheology

Ice is a non-Newtonian fluid, which deforms under its own weight by compressing vertically and stretching laterally, causing lateral ice flow. The flow rate, first suggested by Glen (1955), has been empirically found to be a power law relating strain rate $\dot{\epsilon}$ and stress $\tilde{\sigma}$,

$$\dot{\epsilon} = A\tilde{\sigma}^n,$$

where $A$, $n$ are rheological parameters. This was extended to a floating ice-shelf model by Weertman (1957), and later by Macayeal and Barcilon (1988). From this work, GP03 use a Glen's flow law to describe ice velocity $v(\theta, t)$ in terms of rheological parameters:

$$\nabla \cdot v(\theta, t) = \mu^n h(\theta, t)^n, \tag{A1}$$

with $n = 3$, where $\mu$ is a viscosity parameter, given by Weertman (1957):

$$\mu = \frac{1}{4}\rho_i g \left(1 - \frac{\rho_i}{\rho_w}\right)\overline{A^{1/n}}.$$

Discussion Paper | Discussion Paper | Discussion Paper | Discussion Paper |

**ESDD**

doi:10.5194/esd-2015-84

**Ocean heat transport in Snowball Earth scenario**

D. Comeau et al.

Here $\rho_i$, $\rho_w$ are the densities of ice and water, $g$ is the acceleration due to gravity, and $\overline{A}$ is the depth-averaged Glen's flow law parameter (Sanderson, 1979; Goodman and Pierrehumbert, 2003; Li and Pierrehumbert, 2011), given by (suppressing dependence on colatitude $\theta$)

$$\overline{A} = \frac{1}{h} \int_{-h}^{0} A_0 \exp\left(\frac{-Q}{RT(z)}\right) dz.$$

$R$ is the gas constant, $A_0$ and $Q$ are parameters split by a temperature boundary (as in Barnes et al., 1971; Goodman and Pierrehumbert, 2003; Li and Pierrehumbert, 2011) given in Table 2, and $T(z)$ is the vertical temperature profile through the ice. In the event that ice reaches the Equator, we follow the treatment of GP03 to bring the velocity at the Equator to zero through a balance of a back-pressure term from the force of ice colliding with ice from the other hemisphere (assuming symmetry of Northern and Southern Hemispheres). They found the appropriate version of Eq. (A1) in this case is

$$\frac{\partial v(\theta,t)}{\partial \theta} + v(\theta,t)\cot\theta = r_E \mu^n \left(h(\theta,t) - \frac{b}{h(\theta,t)}\right)^n,$$

where the back-pressure constant $b$ satisfies

$$0 = v(\theta = 90°) = r_E \int_{0}^{2\pi} \mu^n \left(h(\theta,t) - \frac{b}{h(\theta,t)}\right)^n \sin\theta \, d\theta, \tag{A2}$$

which we solve by Newton's method.

*Acknowledgements.* Funding for this work was received from GoMRI/BP and from NSF DMS grant 0304890. J. M. Restrepo also wishes to thank the American Institute of Mathematics, an NSF-funded institute where some of this research was done. The authors also wish to thank Raymond Pierrehumbert and Dorian Abbot for stimulating conversations.

---

Discussion Paper | Discussion Paper | Discussion Paper | Discussion Paper | Discussion Paper |

**ESDD**

doi:10.5194/esd-2015-84

**Ocean heat transport in Snowball Earth scenario**

D. Comeau et al.

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

**Table 1.** Summary of results. Asterisk ($^*$) indicates simulations with insolation at 94 % of present day values. Negative circulation values indicate surface poleward flow.

| Simulation | Emissivity | Circulation | Heat transport |
|---|---|---|---|
| Ice free | $\varepsilon = 0.6$ | −28.8 Sv | $8.1 \times 10^{-1}$ PW |
| Partial ice | $\varepsilon = 0.7$ | −26.6 Sv | $6.7 \times 10^{-1}$ PW |
| Near global ice$^*$ | $\varepsilon = 0.84$ | 2.7 Sv | $1.9 \times 10^{-3}$ PW |
| Global ice$^*$ | $\varepsilon = 0.85$ | 4.9 Sv | $3.3 \times 10^{-4}$ PW |

**Table 2.** Physical parameters used in simulations. See text for details on model initialization and settings of unconstrained parameters.

| Parameter | Symbol | Units | Value |
|---|---|---|---|
| Hemispheric extent | $\ell$ | $\theta$ | $\pi/2$ |
| Extent of Box 2 and 3 | $\zeta$ | $\theta$ | $\pi/4$ |
| Depth of Box 1 and 2 | $d_u$ | m | 200 |
| Depth of Box 3 and 4 | $d_l$ | m | 3000 |
| Volume of Box 1 | $V_1$ | m$^3$ | $2.83 \times 10^{16}$ |
| Volume of Box 2 | $V_2$ | m$^3$ | $1.17 \times 10^{16}$ |
| Volume of Box 3 | $V_3$ | m$^3$ | $1.76 \times 10^{17}$ |
| Volume of Box 4 | $V_4$ | m$^3$ | $4.25 \times 10^{17}$ |
| Earth's radius | $r_E$ | m | $6.371 \times 10^7$ |
| Hydraulic constant | $k_0$ | m$^6$ kg$^{-1}$ s$^{-1}$ | $7.8 \times 10^7$ |
| Reference density | $\rho_0$ | kg m$^{-3}$ | 1027 |
| Reference salinity | $S_0$ | psu | 35 |
| Reference temperature | $T_0$ | K | 283 |
| Salinity exp. coefficient | $\beta_S$ | psu$^{-1}$ | $7.61 \times 10^{-4}$ |
| Temperature exp. coefficient | $\beta_T$ | K$^{-1}$ | $1.668 \times 10^{-4}$ |
| Ocean albedo | $\alpha_w$ | – | 0.32 |
| Sea ice albedo | $\alpha_i$ | – | 0.62 |
| Ocean water heat capacity | $c_w$ | J kg$^{-1}$ K$^{-1}$ | 3996 |
| Sea ice heat capacity | $c_i$ | J kg$^{-1}$ K$^{-1}$ | 2100 |
| Ocean water density | $\rho_w$ | kg m$^{-3}$ | 1027 |
| Ice density | $\rho_i$ | kg m$^{-3}$ | 917 |
| Ocean water conductivity | $\kappa_w$ | W m$^{-1}$ K$^{-1}$ | 0.575 |
| Sea ice conductivity | $\kappa_i$ | W m$^{-1}$ K$^{-1}$ | 2.5 |
| Sea ice latent heat | $L$ | J kg$^{-1}$ | $3.34 \times 10^5$ |
| Freezing temperature | $T_f$ | K | 271.2 |
| Stefan–Boltzmann constant | $\sigma$ | J m$^{-2}$ s K$^{-4}$ | $5.6704 \times 10^{-8}$ |
| Ice/ocean boundary layer | $D$ | m | 0.05 |
| Ice viscosity parameter | $A_0$ | Pa$^{-3}$ s$^{-1}$ | $3.61 \times 10^{-13}$, $T < 263.15$; $1.734 \times 10^{-13}$, $T > 263.15$ |
| Ice viscosity parameter | $Q$ | J mol$^{-1}$ | $60 \times 10^3$, $T < 263.15$; $139 \times 10^3$, $T > 263.15$. |
| Gas constant | $R$ | J K$^{-1}$ mol$^{-1}$ | 8.31446 |
| Acceleration due to gravity | $g$ | m s$^{-2}$ | 9.8 |
| Glen's flow law exponent | $n$ | – | 3 |
| Geothermal heat forcing | $F_g$ | W m$^{-2}$ | 0.05 |

Discussion Paper | Discussion Paper | Discussion Paper | Discussion Paper |

**ESDD**

doi:10.5194/esd-2015-84

**Ocean heat transport in Snowball Earth scenario**

D. Comeau et al.

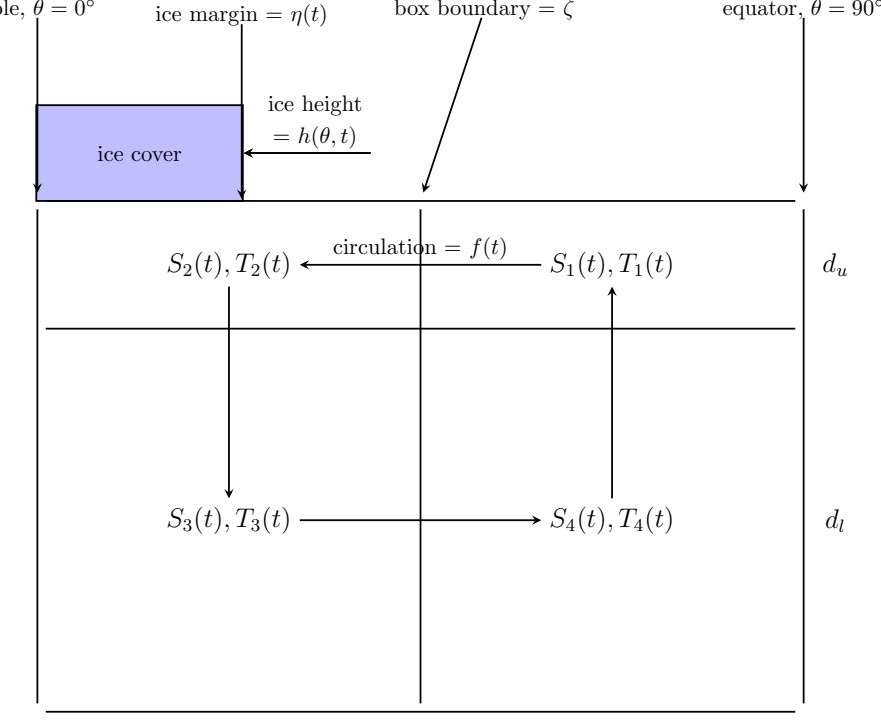

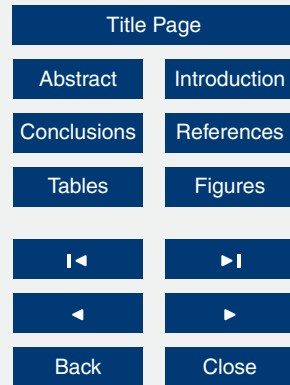

**Figure 1.** Hemispheric four-box arrangement. Boxes 1 and 2 are the surface ocean boxes of depth $d_u$, and Boxes 3 and 4 are the deep ocean boxes of depth $d_l$. The water in Box $j$ has (well-mixed) salinity $S_j$ and temperature $T_j$. The boundary between polar and equatorial boxes is at colatitude $\zeta$. Ice cover sits atop the surface boxes with height $h$ and the ice margin at $\eta$. The arrows between the boxes represent density driven circulation $f$.



Discussion Paper | Discussion Paper | Discussion Paper | Discussion Paper |

**ESDD**

doi:10.5194/esd-2015-84

**Ocean heat transport in Snowball Earth scenario**

D. Comeau et al.

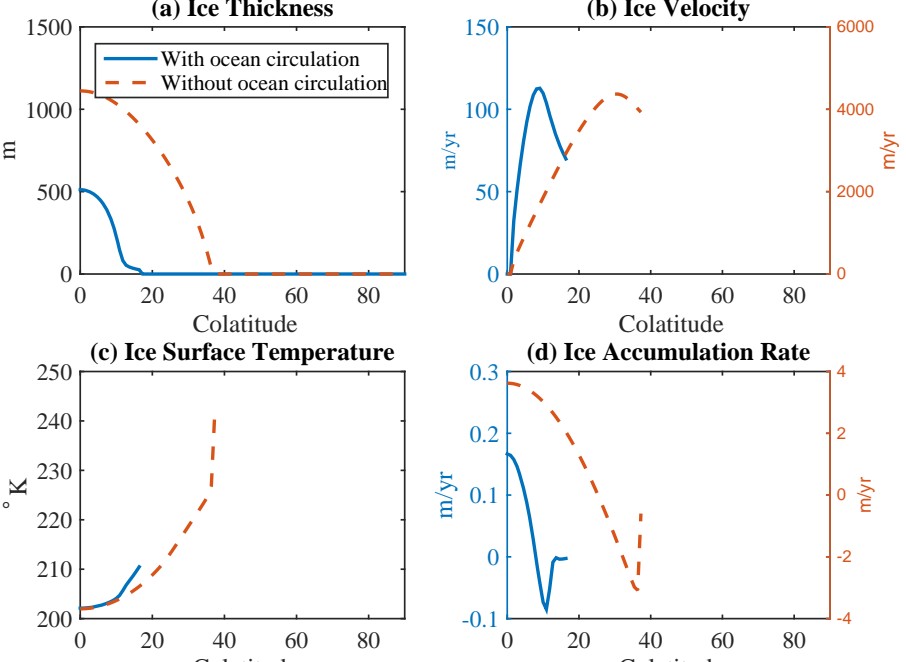

**Figure 2. (a)** Ice thickness $h$ and **(b)** meridional velocity $v$ profiles in partial ice cover scenario with effective emissivity $\varepsilon = 0.7$, with and without ocean circulation. Oceanic transport is seen to dramatically affect the ice predictions. **(c)** Ice surface temperature $T_s$ and **(d)** ice basal accumulation rate in partial ice cover scenario with effective emissivity $\varepsilon = 0.7$, with and without ocean circulation. The accumulation term stabilizes the ice margin, and the model produces reasonable values of ice surface temperatures. $M$ is positive over the region of ice cover indicating net accumulation of ice, and negative values of $M$ beyond the ice margin indicate any ice present would be melted.

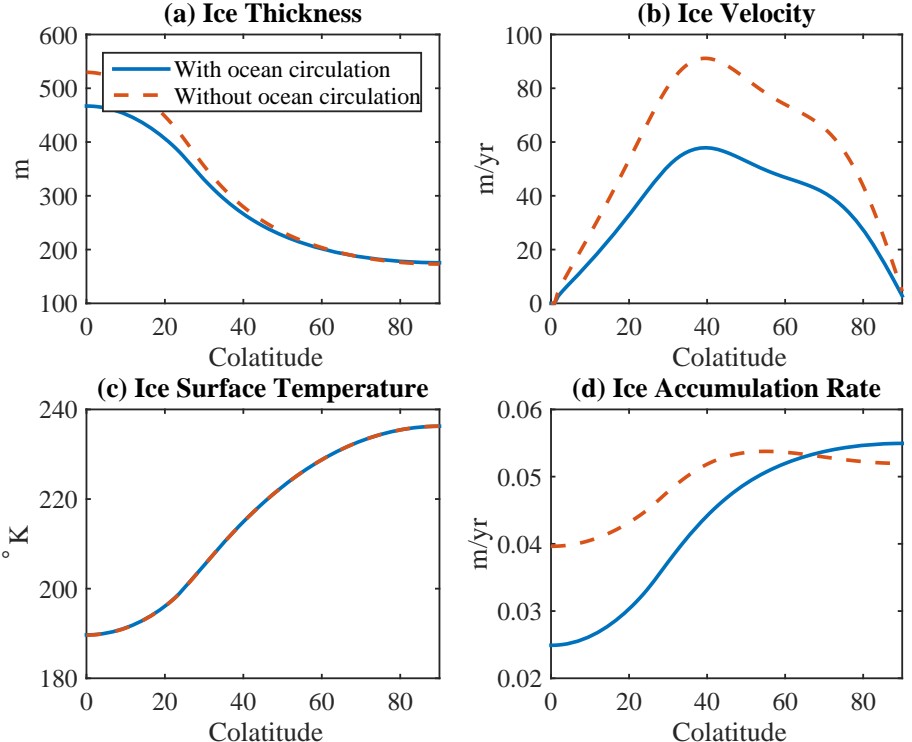

**Figure 3. (a)** Ice thickness $h$ and **(b)** velocity $v$ profiles with 94 % insolation and effective emissivity $\varepsilon = 0.85$, the Snowball Earth scenario. **(c)** Ice surface temperature $T_s$, which is seen to not be impacted by ocean circulation. **(d)** The ice accumulation term, while positive everywhere, has stabilized to small enough values that the profiles only change gradually after around 5000 years.

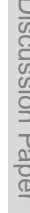

**ESDD**

doi:10.5194/esd-2015-84

**Ocean heat transport in Snowball Earth scenario**

D. Comeau et al.

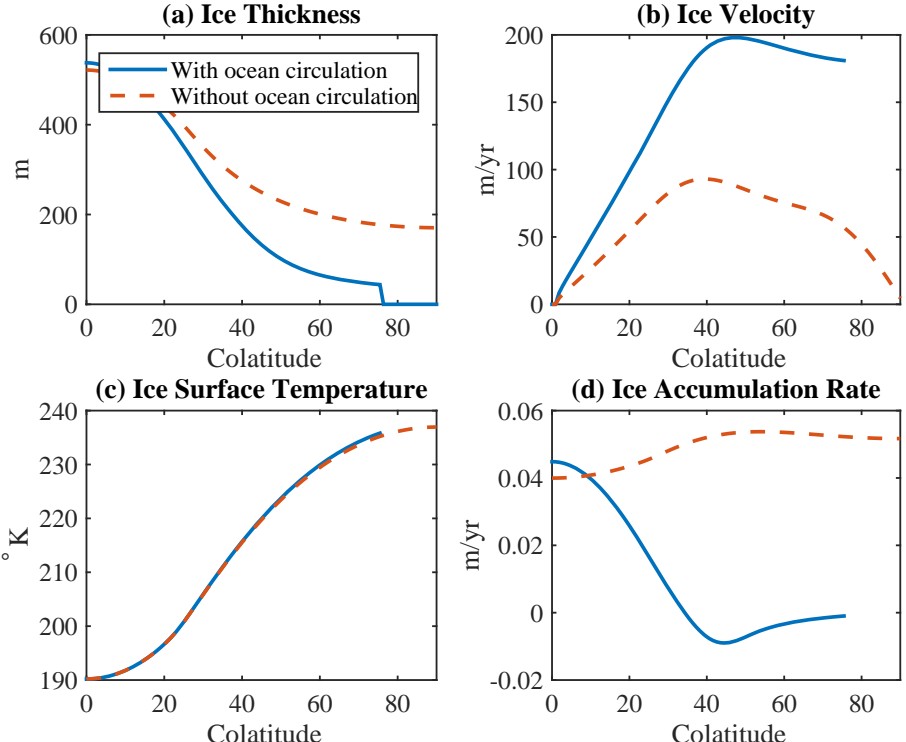

**Figure 4. (a)** Ice thickness $h$ and **(b)** velocity $v$ profiles with 94 % insolation and $\varepsilon = 0.84$, with and without ocean circulation. We note neglecting oceanic heat transport leads to drastically different global climate states. **(c)** Ice surface temperature $T_s$ and **(d)** ice basal accumulation $M$, which without oceanic heat transport is positive everywhere.

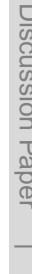

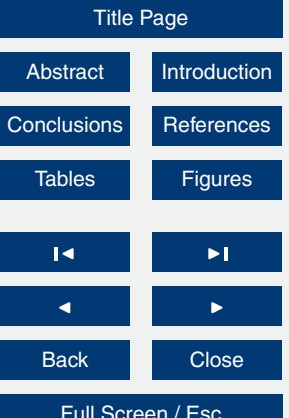

**ESDD**

doi:10.5194/esd-2015-84

**Ocean heat transport in Snowball Earth scenario**

D. Comeau et al.

Full Screen / Esc

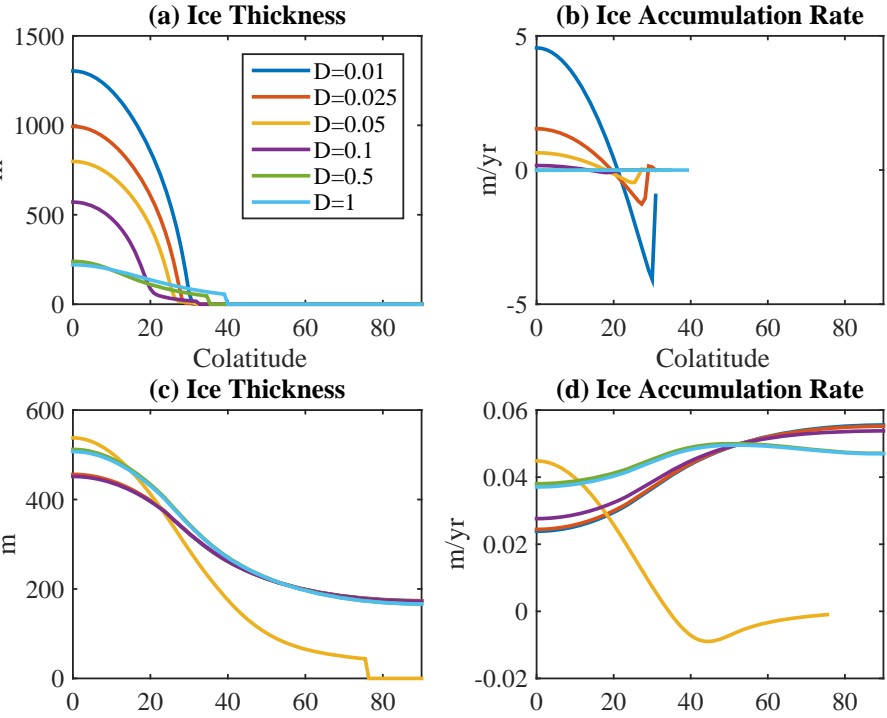

**Figure 5.** Effect of D parameter on equilibrium climate state, small ice cap regime ($\varepsilon = 0.75$, top), and large ice cap regime ($\varepsilon = 0.84$ and 94 % insolation, bottom). There is particularly strong response in the resulting melting/accumulation term which was used to constrain the value of $D$. Furthermore, the instability near the large ice cap threshold is reflected in nearby values of $D$ to the default $D = 0.05$ resulting in global ice cover for $\varepsilon = 0.84$.

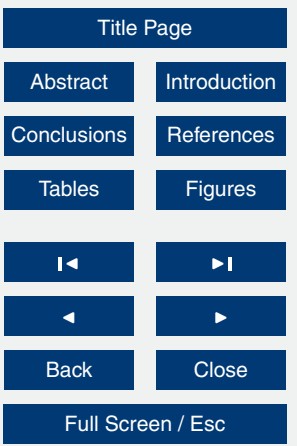

Discussion Paper | Discussion Paper | Discussion Paper | Discussion Paper

**ESDD**

doi:10.5194/esd-2015-84

**Ocean heat transport in Snowball Earth scenario**

D. Comeau et al.



Discussion Paper | Discussion Paper | Discussion Paper | Discussion Paper

**ESDD**

doi:10.5194/esd-2015-84

**Ocean heat transport in Snowball Earth scenario**

D. Comeau et al.

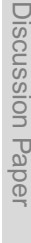



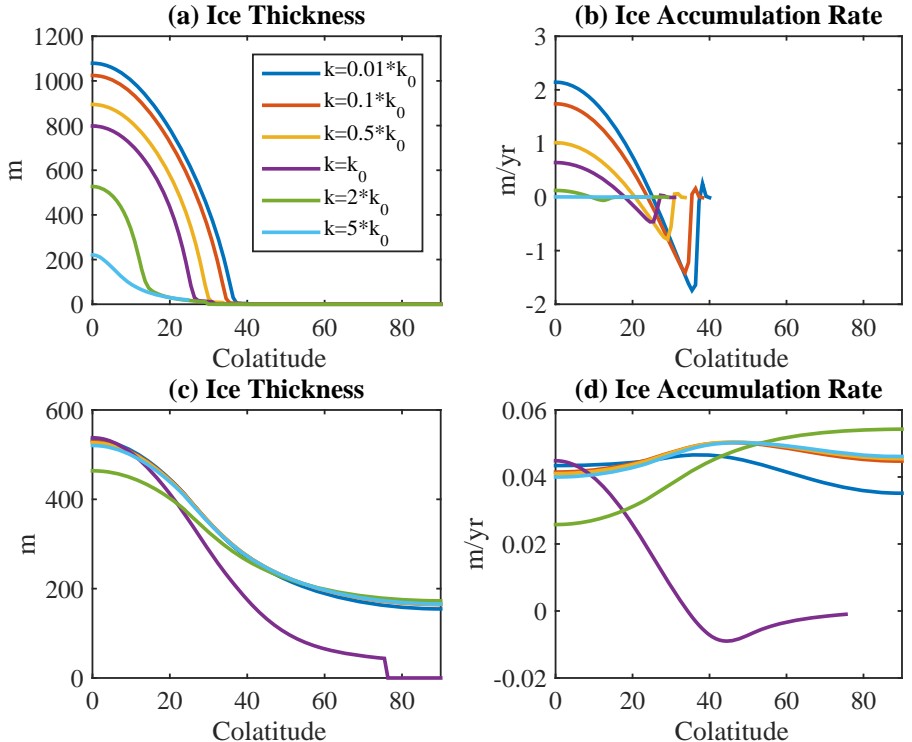

**Figure 6.** Effect of $k_0$ parameter on small ice cap regime (top) and large ice cap regime (bottom). Smaller circulation constants reduce heat transport from the tropics to the poles, thereby facilitating ice growth. The large ice capt instability is again reflected in the sensitivity to changes to the circulation constant, with the shown deviations resulting in Snowball Earth.

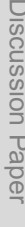

**ESDD**

doi:10.5194/esd-2015-84

**Ocean heat transport in Snowball Earth scenario**

D. Comeau et al.

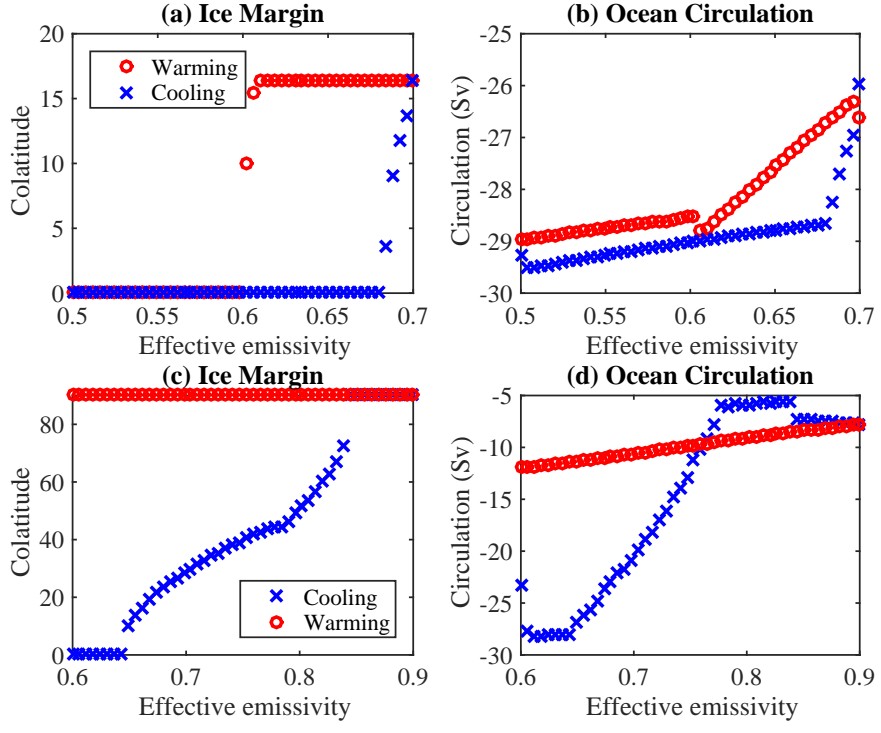

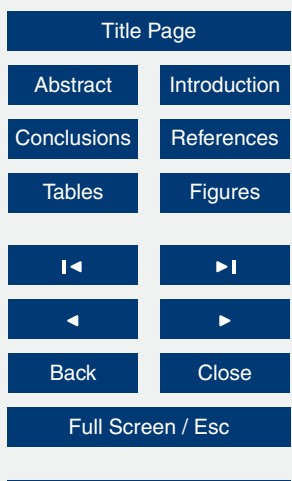

**Figure 7.** Hysteresis experiments, radiative forcing. Top, small ice cap regime: forcing changes in $\varepsilon$, warming first ($\varepsilon = 0.7 \mapsto 0.5$), then cooling ($\varepsilon = 0.5 \mapsto 0.7$). Resulting ice margin (left) and circulation (right) are shown, demonstrating a hysteresis loop in the ice margin. Bottom, large ice cap regime: with insolation set to 94 % present values, forcing changes in $\varepsilon$, cooling first to achieve Snowball Earth ($\varepsilon = 0.6 \mapsto 0.9$), then subsequent warming ($\varepsilon = 0.9 \mapsto 0.6$). Strong hysteresis loop is seen in the ice margin, as the model is unable to escape Snowball Earth even as radiative forcing is raised to levels associated to ice free states, although strong enough increases in radiative forcing will eventually melt ice away from global ice cover. Circulation remains poleward, ever increasing through the Snowball Earth state manual radiative forcing increase.

**ESDD**

doi:10.5194/esd-2015-84

**Ocean heat transport in Snowball Earth scenario**

D. Comeau et al.

Discussion Paper | Discussion Paper | Discussion Paper | Discussion Paper | Discussion Paper |

**ESDD**

doi:10.5194/esd-2015-84

**Ocean heat transport in Snowball Earth scenario**

D. Comeau et al.

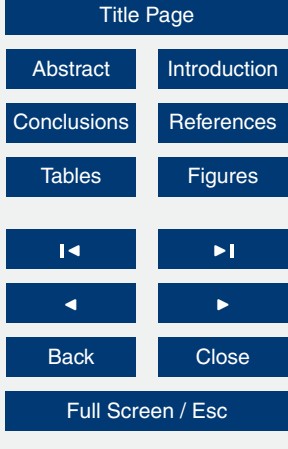

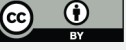

**(a) Ice Margin**

**(b) Ocean Circulation**

**(c) Ice Margin**

**(d) Ocean Circulation**

**Figure 8.** Hysteresis experiments, circulation strength constant. Top, small ice cap regime: forcing changes in $k$, weakening first ($k = k_0 \mapsto 0$), then strengthening ($k = 0 \mapsto k_0$). Bottom, large ice cap regime: forcing changes in $k$, weakening first ($k = k_0 \mapsto 0$), then strengthening back to the default circulation constant ($k = 0 \mapsto k_0$). In the small ice cap experiment, while the ice margin does not melt back from its furthest extent, the resulting ocean circulation actually remains in line as the strength constant is dialed down to 0 and back up again. In the large ice regime, weakening the circulation causes a collapse to Snowball Earth, which is then inescapable as manually increasing the circulation constant only furthers the equatorward circulation.

Discussion Paper | Discussion Paper | Discussion Paper | Discussion Paper | Discussion Paper |

**ESDD**

doi:10.5194/esd-2015-84

**Ocean heat transport in Snowball Earth scenario**

D. Comeau et al.