# Peer review of "A conceptual model of oceanic heat transport in the Snowball Earth scenario"

_Earth System Dynamics, 2015_

## Referee Comment (RC1) · Anonymous Referee #1 · 22 Mar 2016

Review of "A conceptual model of oceanic heat transport in the Snowball Earth scenario" by D. Comeau, D. A. Kurtze and J. M. Restrepo

The authors use a low-order climate model to study the role of the ocean circulation and ocean heat transport for the initiation of hard Snowball Earth episodes (i.e., global sea-ice cover). Besides the investigation of the large-ice cap instability associated with a Snowball Earth, the authors further study the small-ice cap instability. To this end, they develop a simplified coupled atmosphere-ocean-sea-ice model in which the radiative effect of the atmosphere on the surface temperature is represented by a prescribed effective emissivity and the ocean is represented by four boxes with heat transport between them. The ice representation is the most complex part of the model as it includes flow of thick sea ice under its own weight (i.e., sea glaciers). The authors use the model to demonstrate that 1) ocean heat transport works against Snowball Earth

initiation since a Snowball Earth results when they shut off the ocean circulation, 2) the heat exchange at the ocean-ice interface has a strong impact on the ice edge in this model, and 3) the Snowball ocean circulation can either be directed poleward or equatorward. I find the paper very interesting, well suited for Earth System Dynamics and a valuable contribution to the literature on Snowball Earth climate dynamics. I do have a couple of suggestions and comments though that I hope the authors will be able to address before publication.

Major comments:

1. Atmospheric component of the model: there seems to be no representation of atmospheric heat transport, in contrast to the classic EBMs of Budyko and Sellers. Indeed, the Budyko and Sellers models only have multiple stable states because of atmospheric heat transport (Held and Suarez, 1974, Simple Albedo Feedback Models of the Ice Caps; Fig. 3). This seems worth pointing out because it implies that the bistability found in the model used here is different from the stability found in atmosphere-ice EBMs without ocean heat transport. It also makes me wonder to what extent the solutions would differ if a representation of atmospheric transport was included. I.e., would there be more equilibrium solutions, less, or the same? Answering this might require much work and might go beyond what is possible in the revision, but I would appreciate if the authors devoted some discussion to these points.

2. The model is hemispherically symmetric and has no cross-equatorial ocean flow, but much of the ocean heat transport is achieved by the cross-equatorial AMOC in the present-day climate, with upwelling in the Southern Ocean and downwelling in the North Atlantic. So are comparisons between the Sv of the model's ocean circulation and present-day observations really meaningful?

3. Conductive heat flux at ocean-ice interface, Eqs. 3 and 4: the model includes heat conduction proportional to the temperature difference between the ice freezing temperature and the ocean temperature. But for salinities greater 24.7 the density

maximum of sea water is at its freezing point and the formation of sea ice at the surface must therefore be preceded by convection due to stability reasons (e.g., Washington and Parkinson 2005; Voigt and Marotzke, Climate Dynamics 2010). So shouldn't this term then not always be zero, since I expect salinities are above 24.7 psu. Apparently it is not, as is shown by the importance of that term, but it remains unclear to me why. Maybe for a partially ice covered box the ocean temperature will be above freezing temperature since otherwise the box was completely ice covered, but then it seems physically dubious to use the mean ocean temperature to parametrize the ocean-ice heat flux as this relies on the temperature of the ocean region where there is no sea ice, and hence no ocean-ice flux. How does this affect the result on the importance of the ocean-ice flux?

4. Global-mean ice-accumulation rates: In equilibrium the global-mean ice-accumulation rate must be zero. In Fig. 2d, the global-mean of the red dashed line seems to be positive if looking at the blue axes, but now that I notice the red axes and that it might indeed be zero. So the blue and the red axes should have the same zero position. Related to this, the global-mean accumulation rate in a Snowball Earth state must be zero, but in Figs. 3d, 4d it is positive everywhere, and so the ice is still growing and the model is not yet in equilibrium. I suggest to run the model into true equilibrium. The final ice thickness will be set by the value of the geothermal heat flux.

5. It's very interesting to see that the model can sustain a stable ice edge at 10-15 deg of the equator. This is similar to the Jormungand state (Abbot, Voigt, Koll, JGR 2011; Voigt and Abbot, Climate of the Past 2012) but must be for a different reason. In Abbot et al. and Voigt and Abbot, the stable low-latitude ice edge was made possible by the low albedo of snow-free sea ice and ocean transport was not important. Here, in contrast, the stability must be due to ocean transport, and a more detailed discussion of the physics that are involved in the stabilization of the ice edge by the ocean transport would be desirable.

6. In the abstract the authors state that the presence of ocean heat transport works

against Snowball initiation. But the impact seems to be non-monotonic according to Fig. 5c: I.e., an intermediate circulation strength (D=0.05) prevents global ice cover, but for both a lower and a higher value of D and thus a weaker and a stronger circulation the model falls into a Snowball.

7. The fact that the Snowball ocean can flow in both directions is interesting, and while it is mentioned in the text I feel that having a figure illustrating this should help. Maybe a more general schematic that describes the connection between ocean circulation and ice edge in the different climates?

Minor comments:

1. Use of colatitude instead of latitude: I suggest the authors change to latitude as this is more familiar for climate scientists, and when looking at the plots I had to sometimes remind myself that 0 degree is not the equator but the pole.

2. I suggest to integrate Eq. 7 with respect to longitude as the model is zonally-symmetric

3. page 13, line 17: long wave –> longwave

4. page 16: longitudinal domain –> latitudinal domain?

5. page 16: for documentation purposes I recommend including a plot for the control climate described here

6. Heading of Section 4.2: Circulation constant –> Hydraulic constant?

7. page 23: broken sentence on lines 14-16

8. Fig. 7: include the melting occurring when epsilon=0.4; this is discussed in the text but not shown in the figure

9. Figs. 8 and 9: include arrows to further indicate the direction in which the experiments are performed; also for the values of epsilon and D for which the blue crosses

and red circles overlap I suggest to plot alternating crosses and circles, otherwise it is difficult to see that there are two experiments leading to the same climate

---

## Referee Comment (RC2) · Anonymous Referee #2 · 9 Apr 2016

I love a good box climate model, and the authors' approach to simplifying complex Snowball climate dynamics down to its fundamentals is admirable. However, the authors have made two serious errors in their analysis, one conceptual and one mathematical:

1) No model of global oceanic heat transport can neglect atmospheric heat transport: the latter is about six times as large as the former in the modern climate.

2) The authors' integrals calculating top-of-atmosphere heat fluxes and geothermal heat fluxes in equations 3-6 are missing a factor of 2 pi.

In a surprising coincidence, these errors cancel each other out, leading to reasonable values for the model's oceanic heat transport and temperature gradient. See the discussion below.

[Figure]

I have two second-tier concerns:

3) I don't think the model has reached equilibrium, which calls into question the authors' results.

4) Like atmospheric heat transport, atmospheric water transport should not be ignored.

Unfortunately, there's not a simple fix which would allow the paper to be published with revisions. Solving problem #3 is easy. Fixing the math error in #2 is simple, but would result in a model climate which is wildly different from reality, because of the conceptual error #1. Fixing that would require major changes to the model, and a total reanalysis of the results.

I rush to add that I really like the philosophy and goals of this paper, and when the authors have addressed the flaws I've outlined here, I would love to review the new manuscript.

I discuss these problems in more detail below, along with additional minor comments.

=================================================

1) Neglect of atmospheric heat transport.

For a given surface temperature, the net heating or cooling of each latitude is fixed by the top-of-atmosphere radiation balance (Trenberth and Caron 2001, Figure 1). Any imbalance in TOA radiation in a given latitude band must be compensated by net heat transport into or out of that band (TC01, Figure 2). This could be achieved by atmosphere or ocean, but in the present-day climate, only about 1/5 of this is carried by the ocean: the atmosphere does most of the work (TC01, Fig 7).

The authors' ocean model is driven directly by TOA radiation balance integrated over the surface of their box, and the integrand is a good match to observations (TC01 Fig 1). The integral of TOA radiation over the tropical box should amount to 5 petawatts, as in observations, but unlike the real world, in their model all of this must be transported

by the ocean: their ocean's heat transport must be five times greater than reality.

To achieve this, the model's ocean circulation would need to be five times more vigorous, or the tropic/polar temperature difference about five times greater, than is observed. This would make the model's base climate so divergent from reality that their Snowball experiments – which depend crucially on meridional temperature gradient – would be called into question.

However, on Page 16, they note that the model produces quite reasonable ocean circulation and temperature gradients. How can this be? Their integral of TOA radiation is too small by a factor of 2 pi, which nicely cancels out the neglect of the atmosphere in the modern-day climate (Point 2 below).

The atmospheric heat transport could be included in a box model. I would suggest the authors track surface air temperature separately from ocean temperature: determine it diagnostically by a balance between TOA radiation, atmospheric heat transport from one box to the other, and air-sea heat exchange:

TOA = F_atmos + F_airsea

Let flux between atmospheric boxes, and exchange between air and sea be governed by the same sort of mixing dynamics used for the ocean boxes, with exchange constants tuned to match observations. Crucially, there is no time derivative in this equation, so the atmosphere temperatures can be solved algebraically given the temperature of the ocean boxes in each timestep.

The oceans would be forced entirely by F_airsea in such a model, rather than the TOA fluxes: since these are weaker, the ocean circulation would be in line with observations.

It's all quite doable, but it's a different paper than the one the authors have submitted. If they wish to go in this direction, I would be delighted to review the manuscript again in the future.

2) Missing factor in integrals.

The integrals in equations 3-6 are intended to be integrals of a vertical flux (watts per square meter) over the surface area of the box, to give a total flux in watts. They are obtained by multiplying the flux at each colatitude by the area of a thin circular "ribbon" encircling the globe at that colatitude, and then adding up all such ribbons. The north-south height of the ribbon is dy = r_e*dtheta; its east-west length is the circumference of the globe at that colatitude, 2*pi*r_e*sin(theta). The 2*pi factor is missing from these equations.

As a result, the authors' radiative forcing is 6.28 times weaker than it should be!

3) Model not at equilibrium.

I was alarmed by the statement on Page 18: that the ice thickness increases con-tinuously in the fully glaciated model. Eventually, the ice should thicken enough that geothermal heat input to the ocean balances conductive heat loss through the ice shell. Goodman et al (2003) argue that this should happen at an ice shell thickness of 1200 meters, and in their model equilibrium is achieved only after 30,000 years. It's possi-ble the authors' model is not yet at equilbrium: they shows a thickness of just 300 m (and still thickening), achieved after a shorter run of 20,000 years. The authors' model may take even longer to reach equilibrium than Goodman (2003), because their ice boundary layer parameterization may limit the rate of heat transfer.

4) Atmospheric moisture fluxes.

Goodman (2006) finds that about 20% of the thickness of a sea glacier would be com-posed of ice derived from snowfall ("meteoric" rather than "marine" ice.) And that's in a hard snowball climate where precipitation is measured in mm per year! In the partially-glaciated cases considered here, the thickening and flow of the ice sheet due to snowfall probably cannot be neglected. A box-model-style mixing parameterization between atmospheric boxes, with a Clausius-Claperyon dependence on temperature could probably be used here.

==================================================

Minor comments:

Page 4 Line 18: "The circulation itself is driven by wind stress in and near the Southern Ocean" — need citation; note Munk and Wunsch 1998 "Abyssal Recipes II" which argues that the AMOC is driven partly by mechanical mixing from tidal action.

Page 4 Line 20: head—> heat

Page 10: Box numbers are hard to keep straight. I'd suggest labeling the variables like this: rho_ut = upper tropics, rho_dp = deep poles, etc.

Page 14: Can D be estimated or justified from observations of modern ice shelves?

Page 16, line 6: latitudinal, not longitudinal, right?

Equation 7 appears to be valid for theta = latitude, not colatitude. Presumably the authors have fixed this mixup in their code, or they'd get box 1 colder than box 2!

Page 18: Why is the ocean circulation negative (sinking at the equator) in the Snowball simulations? In equilibrium, the sea glacier should be freezing at the pole, and melting at the equator, so brine rejection would lead to high-latitude sinking. Temperature effects should be negligible, since the ocean should be at the freezing point everywhere.

—————————————————————

---

## Author Comment (AC1) · 6 May 2016

We would like to thank Reviewer #1 for the helpful and insightful comments on the manuscript.

Major Comments 1. We certainly appreciate the importance of atmospheric heat transport in the global energy budget and climate system. Our modeling approach for the atmosphere was to simply provide a parameterization to express the net forcing of the atmosphere on the ocean and ice system, which was done through the radiative balance forcing in the effective emissivity parameter \$\epsilon\$. Including atmospheric heat transport explicitly would require additional state variables of temperature at some atmospheric level, and then additional coupling parameterizations to the ocean and ice state variables of interest, which were the primary focus of the study. The Held &

Suarez reference is certainly an appropriate one to include in the discussion of our model's bistability. The response of the ocean circulation in the bistability experiments shown in Figs 7 & 8 also responded with a hysteresis loop (with the exception of Figure 8b), though it is a good point to also depict the associated ocean heat transport in these figures, which can be done in the revision.

2. Our hemispheric setup is similar to that of Griffies & Tziperman 1995, from which we get the reference hydraulic constant k. The comparison is just to state that our results are in a reasonable range, and we explore the model's sensitivity to this parameter governing the circulation strength in Section 4.

3. Convective instability at the ocean ice interface is a small scale process well below our model's resolution, and the role of the boundary layer parameter \$D\$ is precisely to parameterize this unresolved process.

4. This is a valid point, and new figures for the global glaciation experiment at true equilibrium will be revised. The climate state in Figure 3 is changing much slower than its transition towards this state, though clearly not at equilibrium with net ice accumulation everywhere. Running the model further results in net melting in the tropics.

5. The ice edge is stabilized by ice production/melting term (Eq. 10), is sufficiently negative as to overpower any ice advancing through advection. It would certainly be worth an examination of the two competing terms that make up this equation individually perhaps a plot showing the contribution of each, noting what happens at or just before ice edge.

6. Rather than the statement that ocean heat transport works against Snowball initiation, we clarify that an appropriate statement would be poleward heat transport works against Snowball initiation. The non-monotonic impact of D in Figure 5c can be seen as a monotonic influence on ocean circulation as switches direction.

7. Perhaps a similar hysteresis example in large ice cap regime (Fig 7, 8) with \$D\$
varying would illustrate this.

Minor comments In general we agree with the minor comments and will revise accordingly., with the exception of #2. Even though our model is zonally averaged, integrating out longitude in the insolation equation results in a much messier looking expression that seems unnecessary.

---

## Author Comment (AC2) · 6 May 2016

We would like to thank Reviewer #2 for the helpful and insightful comments on the manuscript.

Major Comments 1. We certainly agree that atmospheric heat transport is crucial to the global climate system and energy budget. Our modeling approach was to incorporate the atmosphere as a forcing, rather than a coupled aspect of the model. The air-sea heat flux component is an important aspect of heat transport, but at coarse spatio-temporal scales it is important to understand the basic state of outcomes before adding these fluxes with their very strong dependence on local/resolved dynamics. We believe we are not asserting that the model is not transporting all of the planet's heat through the ocean, as is stated by the reviewer. The atmosphere is a forcing in our model

through $\epsilon$ - heat transport within the atmosphere is not modeled, but is all wrapped up in $\epsilon$ - perhaps having different $\epsilon$s for each box (or even a pole-to-equator profile) could mock up varying atmospheric forcing by latitude, at the cost of introducing another parameter. The atmosphere may well transport six times as much heat - our approach is that the net effect of the atmosphere (including heat transport within the atmosphere) on the ocean / ice coupled system is through the radiative balance via $\epsilon$. The Trenberth and Caron reference is a useful one, particularly Figure 5 for reference ocean heat transport values.

2. We thank the reviewer for pointing this out. This has been corrected in the code, and does not qualitatively change results. All corresponding equations have been modified, and the experiment results from the corrected code will be in the revised manuscript.

4. This is a valid point, and new figures for the global glaciation experiment at true equilibrium will be revised. The climate state in Figure 3 is changing much slower than its transition towards this state, though clearly not at equilibrium with net ice accumulation everywhere. Running the model further results in net melting in the tropics.

4. We were aiming for the simplest model that would capture ocean heat transport in Snowball Earth scenario. We certainly agree that precipitation minus evaporation is missing from the model and would affect the results, though we would expect the main results of our study regarding ice/ocean interaction through heat transfer and its affect on the global climate state to stand.

Minor comments In general we agree with the minor comments and will revise accordingly. We are not aware of observations of modern ice shelves that would assist in estimating $D$, though this would certainly be extremely useful data to have.

---

## Author Response (AR1)

**A conceptual model of oceanic heat transport in a Snowball Earth scenario**

Darin Comeau, Douglas A. Kurtze, Juan M. Restrepo

Center for Atmosphere Ocean Science, Courant Institute of Mathematical Sciences, New York University, New York, NY, USA

The authors thank both reviewers their time in providing for insightful comments and helpful critique on the manuscript.

Reviewer #1:

*The authors use a low-order climate model to study the role of the ocean circulation and ocean heat transport for the initiation of hard Snowball Earth episodes (i.e., global sea- ice cover). Besides the investigation of the large-ice cap instability associated with a Snowball Earth, the authors further study the small-ice cap instability. To this end, they develop a simplified coupled atmosphere-ocean-sea-ice model in which the radiative effect of the atmosphere on the surface temperature is represented by a prescribed effective emissivity and the ocean is represented by four boxes with heat transport between them. The ice representation is the most complex part of the model as it includes flow of thick sea ice under its own weight (i.e., sea glaciers). The authors use the model to demonstrate that 1) ocean heat transport works against Snowball Earth initiation since a Snowball Earth results when they shut off the ocean circulation, 2) the heat exchange at the ocean-ice interface has a strong impact on the ice edge in this model, and 3) the Snowball ocean circulation can either be directed poleward or equatorward. I find the paper very interesting, well suited for Earth System Dynamics and a valuable contribution to the literature on Snowball Earth climate dynamics. I do have a couple of suggestions and comments though that I hope the authors will be able to address before publication.*

Major Comments

1. *Atmospheric component of the model: there seems to be no representation of atmospheric heat transport, in contrast to the classic EBMs of Budyko and Sellers. Indeed, the Budyko and Sellers models only have multiple stable states because of atmospheric heat transport (Held and Suarez, 1974, Simple Albedo Feedback Models of the Ice Caps; Fig. 3). This seems worth pointing out because it implies that the bistability found in the model used here is different from the stability found in atmosphere-ice EBMs without ocean heat transport. It also makes me wonder to what extent the solutions would differ if a representation of atmospheric transport was included. I.e., would there be more equilibrium solutions, less, or the same? Answering this might require much work and might go beyond what is possible in the revision, but I would appreciate if the authors devoted some discussion to these points.*

**Response**: We certainly appreciate the importance of atmospheric heat transport in the global energy budget and climate system. Our modeling approach for the atmosphere was to simply provide a parameterization to express the net forcing of the atmosphere on the ocean and ice system, which was done through the radiative balance forcing in the effective emissivity parameter $\epsilon$. Including atmospheric heat transport explicitly would require additional state variables of temperature at some atmospheric level, and then additional coupling parameterizations to the ocean and ice state variables of interest, which were the primary focus of the study. The Held & Suarez reference is certainly an appropriate one and has been added to the discussion of our model's bistability. The response of the ocean circulation in the bistability experiment shown in Figs 7 also responded with a hysteresis loop. Further discussion/clarity/disclosure on our modeling choice for the atmosphere has been added at the end of Section 2.2.

2. *The model is hemispherically symmetric and has no cross-equatorial ocean flow, but much*

*of the ocean heat transport is achieved by the cross-equatorial AMOC in the present-day climate, with upwelling in the Southern Ocean and downwelling in the North Atlantic. So are comparisons between the Sv of the model?s ocean circulation and present-day observations really meaningful?*

**Response**: Our hemispheric setup is similar to that of Griffies & Tziperman 1995, from which we get the reference hydraulic constant $k$. The comparison is just to state that our results are in a reasonable range, and we explore the model's sensitivity to this parameter governing the circulation strength in Section 4.

3. *Conductive heat flux at ocean-ice interface, Eqs. 3 and 4: the model includes heat conduction proportional to the temperature difference between the ice freezing temperature and the ocean temperature. But for salinities greater 24.7 the density maximum of sea water is at its freezing point and the formation of sea ice at the surface must therefore be preceded by convection due to stability reasons (e.g., Washington and Parkinson 2005; Voigt and Marotzke, Climate Dynamics 2010). So shouldn?t this term then not always be zero, since I expect salinities are above 24.7 psu. Apparently it is not, as is shown by the importance of that term, but it remains unclear to me why. Maybe for a partially ice covered box the ocean temperature will be above freezing temperature since otherwise the box was completely ice covered, but then it seems physically dubious to use the mean ocean temperature to parametrize the ocean-ice heat flux as this relies on the temperature of the ocean region where there is no sea ice, and hence no ocean-ice flux. How does this affect the result on the importance of the ocean-ice flux?*

**Response**: Convective instability at the ocean ice interface is a small scale process well below our model's resolution, and the role of the boundary layer parameter $D$ is precisely to parameterize this unresolved process. As mentioned in the text, the ocean temperature used is a regularized temperature profile $T_{1,2}$ between the two ocean boxes to prevent an artificial temperature jump at the box boundary entering into the equation for ice melting / accumulation, and localize the bulk temperature information, precisely to minimize this effect of relying on ocean temperature in an ice-free area directly impacting ice production. In the end, this is still an approximation / modeling choice that was made to try to keep the model as simple and low-dimensional as possible, and is why we focus on the importance of this parameterization in our modeling results.

4. *Global-mean ice-accumulation rates: In equilibrium the global-mean ice- accumulation rate must be zero. In Fig. 2d, the global-mean of the red dashed line seems to be positive if looking at the blue axes, but now that I notice the red axes and that it might indeed be zero. So the blue and the red axes should have the same zero position. Related to this, the global-mean accumulation rate in a Snowball Earth state must be zero, but in Figs. 3d, 4d it is positive everywhere, and so the ice is still growing and the model is not yet in equilibrium. I suggest to run the model into true equilibrium. The final ice thickness will be set by the value of the geothermal heat flux.*

**Response**: This is a valid point and an oversight on our part, and new figures for the global glaciation experiment at true equilibrium are in the revised manuscript (Figures 3 - 6). The climate state in Figure 3 is changing much slower than its transition towards this state, though clearly not at equilibrium with net ice accumulation everywhere. Running the model further results in net melting in the tropics, as seen in revised Figures 3 & 4.

5. *It?s very interesting to see that the model can sustain a stable ice edge at 10-15 deg of the equator. This is similar to the Jormungand state (Abbot, Voigt, Koll, JGR 2011; Voigt and Abbot, Climate of the Past 2012) but must be for a different reason. In Abbot et al. and Voigt and Abbot, the stable low-latitude ice edge was made possible by the low albedo of snow-free sea ice and ocean transport was not important. Here, in contrast, the stability must be due to ocean transport, and a*

*more detailed discussion of the physics that are involved in the stabilization of the ice edge by the ocean transport would be desirable.*

**Response**: In short, the ice edge is stabilized by ice production/melting term (Eq. 10), is sufficiently negative as to overpower any ice advancing through advection. The role $D$, the parameterization of the ice - ocean energy exchange, plays a key role in determining the location of the ice margin, as seen in Figure 5.

6. *In the abstract the authors state that the presence of ocean heat transport works against Snowball initiation. But the impact seems to be non-monotonic according to Fig. 5c: I.e., an intermediate circulation strength (D=0.05) prevents global ice cover, but for both a lower and a higher value of D and thus a weaker and a stronger circulation the model falls into a Snowball.*

**Response**: Rather than the statement that ocean heat transport works against Snowball initiation, we clarify that an appropriate statement would be *poleward* heat transport works against Snowball initiation.

7. *The fact that the Snowball ocean can flow in both directions is interesting, and while it is mentioned in the text I feel that having a figure illustrating this should help. Maybe a more general schematic that describes the connection between ocean circulation and ice edge in the different climates?*

**Response**: With the revised simulations, in the studied parameter regimes we no longer see bistability of the ocean circulation direction.

Minor comments
1. The latitudinal coordinate $\theta$ has been changed from colatitude to latitude in the text, equations, and figures.
2. We respectfully disagree - integrating out longitude in the insolation equation (Eq. 7) results in a much messier looking expression that seems unnecessary.
3. Typo corrected.
4. Typo corrected.
5. The figures plot variables associated with the ice component, but there is no ice in the control climate referenced. The associated steady state ocean circulation and heat transport are summarized in Table 1.
6. The terms 'circulation constant' and 'hydraulic constant' were used interchangeably - this has been revised to only use 'hydraulic constant'.
7. The broken sentence has been edited out.
9. The symbols have been changed to triangles, pointing in the direction of the forcing change.

Reviewer #2:
*I love a good box climate model, and the authors? approach to simplifying complex Snowball climate dynamics down to its fundamentals is admirable. However, the authors have made two serious errors in their analysis, one conceptual and one mathematical: 1) No model of global oceanic heat transport can neglect atmospheric heat transport: the latter is about six times as large as the former in the modern climate. 2) The authors? integrals calculating top-of-atmosphere heat fluxes and geothermal heat fluxes in equations 3-6 are missing a factor of 2 pi. In a surprising coincidence, these errors cancel each other out, leading to reasonable values for the model?s oceanic heat transport and temperature gradient. See the discussion below. I have two second-tier concerns: 3) I don?t*

*think the model has reached equilibrium, which calls into question the authors? results. 4) Like atmospheric heat transport, atmospheric water transport should not be ignored. Unfortunately, there?s not a simple fix which would allow the paper to be published with revisions. Solving problem #3 is easy. Fixing the math error in #2 is simple, but would result in a model climate which is wildly different from reality, because of the conceptual error #1. Fixing that would require major changes to the model, and a total reanalysis of the results. I rush to add that I really like the philosophy and goals of this paper, and when the authors have addressed the flaws I?ve outlined here, I would love to review the new manuscript. I discuss these problems in more detail below, along with additional minor comments.*

*Major Comments 1. 1) Neglect of atmospheric heat transport. For a given surface tempera-ture, the net heating or cooling of each latitude is fixed by the top-of-atmosphere radiation balance (Trenberth and Caron 2001, Figure 1). Any imbalance in TOA radiation in a given latitude band must be compensated by net heat transport into or out of that band (TC01, Figure 2). This could be achieved by atmosphere or ocean, but in the present-day climate, only about 1/5 of this is carried by the ocean: the atmosphere does most of the work (TC01, Fig 7). The authors? ocean model is driven directly by TOA radiation balance integrated over the surface of their box, and the integrand is a good match to observations (TC01 Fig 1). The integral of TOA radiation over the tropical box should amount to 5 petawatts, as in observations, but unlike the real world, in their model all of this must be transported by the ocean: their ocean?s heat transport must be five times greater than reality. To achieve this, the model?s ocean circulation would need to be five times more vigorous, or the tropic/polar temperature difference about five times greater, than is observed. This would make the model?s base climate so divergent from reality that their Snowball experiments ? which depend crucially on meridional temperature gradient ? would be called into question. However, on Page 16, they note that the model produces quite reasonable ocean circulation and temperature gradients. How can this be? Their integral of TOA radiation is too small by a factor of 2 pi, which nicely cancels out the neglect of the atmosphere in the modern-day climate (Point 2 below). The atmospheric heat transport could be included in a box model. I would suggest the authors track surface air temperature separately from ocean temperature: determine it diagnostically by a balance between TOA radiation, atmospheric heat transport from one box to the other, and air-sea heat exchange: $TOA = F_atmos + F_airsea$ Let flux between atmospheric boxes, and exchange between air and sea be governed by the same sort of mixing dynamics used for the ocean boxes, with exchange constants tuned to match observations. Crucially, there is no time derivative in this equation, so the atmosphere temperatures can be solved algebraically given the temperature of the ocean boxes in each timestep. The oceans would be forced entirely by $F_airsea$ in such a model, rather than the TOA fluxes: since these are weaker, the ocean circulation would be in line with observations. It?s all quite doable, but it?s a different paper than the one the authors have submitted. If they wish to go in this direction, I would be delighted to review the manuscript again in the future.*

**Response**: We certainly agree that atmospheric heat transport is crucial to the global climate system and energy budget. Our modeling approach was to incorporate the atmosphere as a forcing, rather than a coupled aspect of the model. The air-sea heat flux component is an important aspect of heat transport, but at coarse spatio-temporal scales it is important to understand the basic state of outcomes before adding these fluxes with their very strong dependence on local/resolved dynamics. We believe we are not asserting that the model is not transporting all of the planet's heat through the ocean, as is stated by the reviewer. The atmosphere is a forcing in our model through $\epsilon$ - heat transport within the atmosphere is not modeled, but is all wrapped up in $\epsilon$ - perhaps having different $\epsilon$s for each box (or even a pole-to-equator profile) could mock up varying atmospheric forcing by latitude, at the cost of introducing another parameter.

We believe we are not asserting that the model is not transporting all of the planet's heat through

the ocean, as is stated, and think this sentiment is key to the reviewer's discomfort with our lack of an explicit, dynamic atmosphere. The atmosphere is a forcing in our model through $\epsilon$ - heat transport within the atmosphere is not modeled, but is all wrapped up in $\epsilon$ - perhaps having different $\epsilon$s for each box (or even a pole-to-equator profile) could mock up varying atmospheric forcing by latitude, at the cost of introducing another parameter. The atmosphere may well transport six times as much heat - our approach is that the net effect of the atmosphere (including heat transport within the atmosphere) on the ocean / ice coupled system is through the radiative balance via $\epsilon$. Further discussion/clarity/disclosure on our modeling choice for the atmosphere has been added at the end of Section 2.2.

We thank the reviewer for the useful Trenberth and Caron reference, particularly Figure 5 for reference ocean heat transport values.

2. *Missing factor in integrals. The integrals in equations 3-6 are intended to be integrals of a vertical flux (watts per square meter) over the surface area of the box, to give a total flux in watts. They are obtained by multiplying the flux at each colatitude by the area of a thin circular "ribbon" encircling the globe at that colatitude, and then adding up all such ribbons. The north-south height of the ribbon is $dy = r_e * dtheta$; its east-west length is the circumference of the globe at that colatitude, $2 * pi * r_e * sin(theta)$. The $2 * pi$ factor is missing from these equations. As a result, the authors? radiative forcing is 6.28 times weaker than it should be!*

**Response**: We thank the reviewer for pointing out this glaring missing factor. We have corrected our code and manuscript with the $2\pi$ integral factor, and have rerun all simulations to provide revised all results in the manuscript.

3. *Model not at equilibrium. I was alarmed by the statement on Page 18: that the ice thickness increases continuously in the fully glaciated model. Eventually, the ice should thicken enough that geothermal heat input to the ocean balances conductive heat loss through the ice shell. Goodman et al (2003) argue that this should happen at an ice shell thickness of 1200 meters, and in their model equilibrium is achieved only after 30,000 years. It?s possible the authors? model is not yet at equilbrium: they shows a thickness of just 300 m (and still thickening), achieved after a shorter run of 20,000 years. The authors? model may take even longer to reach equilibrium than Goodman (2003), because their ice boundary layer parameterization may limit the rate of heat transfer.*

**Response**: This is a valid point also raised by Reviewer #1. The climate state in Figure 3 is changing much slower than its transition towards this state, though clearly not at equilibrium with net ice accumulation everywhere. We have provided revised simulations with model for Snowball Earth in equilibrium after 500,000 years. The system changed very slowly after global glaciation was reached, gradually accumulating ice until a region of net melting formed in lower latitudes to balance polar ice production. Of course, in comparative runs with ocean circulation turned off, equilibrium could not be reached as geothermal heat flux had no contact with the ice layer, and ice does (slowly) continue to accumulate with no mechanism in place to allow for melting.

4. *Atmospheric moisture fluxes. Goodman (2006) finds that about 20% of the thickness of a sea glacier would be composed of ice derived from snowfall ("meteoric" rather than "marine" ice.) And that?s in a hard snowball climate where precipitation is measured in mm per year! In the partially-glaciated cases considered here, the thickening and flow of the ice sheet due to snowfall probably cannot be neglected. A box-model-style mixing parameterization between atmospheric boxes, with a Clausius-Claperyon dependence on temperature could probably be used here.*

**Response**: We were aiming for the simplest model that would capture ocean heat transport in Snowball Earth scenario, and the addition of atmospheric moisture flux would have increased complexity beyond the scope of the study. We certainly agree that precipitation minus evaporation

is missing from the model and would quantitatively affect the results, though we would expect the main results of our study regarding ice/ocean interaction through heat transfer and its affect on the global climate state to stand. Furthermore the lack of atmospheric moisture was mentioned at the end of the original manuscript as an area for future work.

Minor comments

1. The note and reference has been added to the text.

2. Typo corrected.

3. The variables related to box quantities have been relabeled from box number to $ut, up, dp, dt$ for upper tropic (Box 1), upper polar (Box 2), deep polar (Box 3), and deep tropic (Box 4), respectively.

4. While recent observational basal melt rates under ice shelves are available (notably Rignot et al 2013 (Science), Depoorter et al 2013 (Nature), and Schodlok et al 2015 (JGR: Oceans), the authors are not aware of an observational ocean temperatures under ice shelves, though pursuing this with GCM simulation or reanalysis data to form an estimate of $D$ would be valuable future work.

5. Typo corrected.

6. Correct, the insolation equation is (mistakenly) for $\theta = $ latitude as written. Per another reviewer's request, we have converted $\theta$ from latitude to colatitude everywhere.

7. With the code modifications addressed, we actually now see poleward circulation in the Snowball simulations, consistent with the reviewers comments.

[revised manuscript text omitted]

$$V_2\frac{dS_{up}}{\delta t} = |f|(S_{1ut} - S_{2up}) + S_{2up}2\pi\int_{\min\zeta,\eta}^{90}\frac{\rho_w}{\rho_i}M(\theta)r_E^2\sin\cos\theta\,d\theta,$$

$$V_3\frac{dS_{dp}}{\delta t} = |f|(S_{2up} - S_{3dp}),$$

$$V_4\frac{dS_{dt}}{\delta t} = |f|(S_{3dp} - S_{4dt}). \tag{2}$$

Here $V_j$ is the volume of the $j$ box, and $M(\theta, t)$ is the total production/melting rate of ice, with

230 $M > 0$ corresponding to ice production and $M < 0$ corresponding to melting, described in Section 2.2. The terms involving the circulation rate $f$ correspond to fluxes across the box boundaries. We assume that ice sits atop the surface ocean boxes, and that the mass of ice is much less than the total mass of ocean water. The surface ocean box volumes $V_j$ are kept constant, and any changes to the deep ocean box volumes are negligible. An important assumption is that ice that is formed is fresh-

235 water ice, and as such rejects brine into the ocean. The integral term represents the change in salinity due to net freshwater added/removed through ice melting/production. The bounds of integration represent the portion of each box covered in ice, and $r_E$ is the Earth's radius. The ice component of the model serves as a saline forcing on the ocean box model component. There are similar equations for when ocean circulation is in the reverse direction.

240 In contrast to traditional Stommel box models, rather than using transport equations for the temperature we opt instead for thermal balance equations. The box temperatures change as heat transfers between boxes with the flow $f$, and as it transfers into the box via net radiation or conduction through

overlying ice. Thus for $f < 0$ the temperature equations are

$$c_w V_1 \frac{d(\rho_1 T_1)}{dt}\frac{d(\rho_{ut} T_{ut})}{\delta t} = c_w |f| \left( \rho_{4dt} T_{4dt} - \rho_{1ut} T_{1ut} \right)$$

$$+ \ 2\pi \int_{\max \zeta, \eta}^{\pi/2} \left( (1 - \alpha_w) F_s(\theta) - \varepsilon \sigma T_{1ut}{}^4 \right) r_E^2 \sin \cos \theta d\theta$$

$$- \ \frac{\kappa_w}{D}(T_{1ut} - T_f) 2\pi \int_{\zeta}^{\max \zeta, \eta} r_E^2 \sin \cos \theta d\theta, \tag{3}$$

$$c_w V_2 \frac{d(\rho_2 T_2)}{dt}\frac{d(\rho_{up} T_{up})}{\delta t} = c_w |f| \left( \rho_{1ut} T_{1ut} - \rho_{2up} T_{2up} \right)$$

$$+ \ 2\pi \int_{\min \zeta, \eta}^{\zeta} \left( (1 - \alpha_w) F_s(\theta) - \varepsilon \sigma T_{2up}{}^4 \right) r_E^2 \sin \cos \theta d\theta$$

$$- \ \frac{\kappa_w}{D}(T_{2up} - T_f) 2\pi \int_{0}^{\min \zeta, \eta} r_E^2 \sin \cos \theta d\theta, \tag{4}$$

$$c_w V_3 \frac{d(\rho_3 T_3)}{dt}\frac{d(\rho_{dp} T_{dp})}{\delta t} = c_w |f|(\rho_{2up} T_{2up} - \rho_{3dp} T_{3dp}) + F_g 2\pi \int_{0}^{\zeta} r_E^2 \sin \cos \theta d\theta, \tag{5}$$

$$c_w V_4 \frac{d(\rho_4 T_4)}{dt}\frac{d(\rho_{dt} T_{dt})}{\delta t} = c_w |f|(\rho_{3dp} T_{3dp} - \rho_{4dt} T_{4dt}) + F_g 2\pi \int_{\zeta}^{\pi/2} r_E^2 \sin \
[revised manuscript text omitted]

| partial ice | $\varepsilon = 0.7$ | -26.6 -44.44 Sv | 6.7e-1 1.83 PW |
| near global ice* | $\varepsilon = 0.84$ $\varepsilon = 0.82$ | 2.7 -16.46 Sv | 1.9e-3 6.27e-4 PW |
| global ice* | $\varepsilon = 0.85$ $\varepsilon = 0.83$ | 4.9 -17.46 Sv | 3.3e-4 3.34e-2 PW |

**Table 2.** Physical parameters used in simulations. See text for details on model initialization and settings of unconstrained parameters.

| Parameter | Symbol | Units | Value |
|---|---|---|---|
| hemispheric extent | $\ell$ | $\theta$ | $\pi/2$ |
| extent of Box 2 and 3 | $\zeta$ | $\theta$ | $\pi/4$ |
| depth of Box 1 and 2 | $d_u$ | $m$ | 200 |
| depth of Box 3 and 4 | $d_l$ | $m$ | 3000 |
| volume of Box 1 | $V_{ut}$ | m³ | $2.83 \times 10^{16}$ |
| volume of Box 2 | $V_{up}$ | m³ | $1.17 \times 10^{16}$ |
| volume of Box 3 | $V_{dp}$ | m³ | $1.76 \times 10^{17}$ |
| volume of Box 4 | $V_{dt}$ | m³ | $4.25 \times 10^{17}$ |
| Earth's radius | $r_E$ | m | $6.371 \times 10^7$ |
| hydraulic constant | $k_0$ | m⁶·kg⁻¹·s⁻¹ | $7.8 \times 10^7$ |
| reference density | $\rho_0$ | kg·m⁻³ | 1027 |
| reference salinity | $S_0$ | psu | 35 |
| reference temperature | $T_0$ | K | 283 |
| salinity exp. coefficient | $\beta_S$ | psu⁻¹ | $7.61 \times 10^{-4}$ |
| temperature exp. coefficient | $\beta_T$ | K⁻¹ | $1.668 \times 10^{-4}$ |
| ocean albedo | $\alpha_w$ | - | 0.32 |
| sea ice albedo | $\alpha_i$ | - | 0.62 |
| ocean water heat capacity | $c_w$ | J·kg⁻¹·K⁻¹ | 3996 |
| sea ice heat capacity | $c_i$ | J·kg⁻¹·K⁻¹ | 2100 |
| ocean water density | $\rho_w$ | kg·m⁻³ | 1027 |
| ice density | $\rho_i$ | kg·m⁻³ | 917 |
| ocean water conductivity | $\kappa_w$ | W·m⁻¹·K⁻¹ | 0.575 |
| sea ice conductivity | $\kappa_i$ | W·m⁻¹·K⁻¹ | 2.5 |
| sea ice latent heat | $L$ | J·kg⁻¹ | $3.34 \times 10^5$ |
| freezing temperature | $T_f$ | K | 271.2 |
| Stefan-Boltzmann constant | $\sigma$ | J·m⁻²·s·K⁻⁴ | $5.6704 \times 10^{-8}$ |
| ice/ocean boundary layer | $D$ | m | 0.05 |
| ice viscosity parameter | $A_0$ | Pa⁻³s⁻¹ | $3.61 \cdot 10^{-13}$  $T < 263.15$; $1.734 \cdot 10^3$  $T > 263.15$. |
| ice viscosity parameter | $Q$ | J·mol⁻¹ | $60 \cdot 10^3$  $T < 263.15$; $139 \cdot 10^3$  $T > 263.15$. |
| gas constant | $R$ | J·K⁻¹·mol⁻¹ | 8.31446 |
| acceleration due to gravity | $g$ | m·s⁻² | 9.8 |
| Glen's flow law exponent | $n$ | - | 3 |
| geothermal heat forcing | $F_g$ | W·m⁻² | 0.05 |

---

## Author Response (AR2)

**A conceptual model of oceanic heat transport in a Snowball Earth scenario**
**Manuscript # esd-2015-84**

Darin Comeau, Douglas A. Kurtze, Juan M. Restrepo
Center for Atmosphere Ocean Science, Courant Institute of Mathematical Sciences, New York University, New York, NY, USA

Technical Corrections:

1. typo in line 175: "with the an albedo"? *Corrected.*

2. the box indices are wrong in a number of equations, instead of ut and up the authors sometimes use st and sp, please check. *Corrected.*

3. in Eq. 2 and following the delta of the time derivative is a Greek delta but should be a Latin d. *Corrected in Eqn (2), (3), and (11).*

4. What is the vertical purple line in Fig. 5d? *This discontinuity is an artifact of the regularization of Eqn (10) as the ice thickness h goes to 0. This has been mentioned in the caption.*

5. Fig. 7: are the latitudes indeed colatitudes? Otherwise I don't understand how the top row can show the small ice cap instability, and the bottom row the large ice cap instability. *Yes, the vertical axis should be colatitude - this has been corrected.*